# Radiative Effect and Climate Impacts of Brown Carbon with the Community Atmosphere Model (CAM5)

Hunter Brown[1], Xiaohong Liu[1,*], Yan Feng[2], Yiquan Jiang[3], Mingxuan Wu[1], Zheng Lu[1], Chenglai Wu[1,4], Shane Murphy[1], and Rudra Pokhrel[1]

[1]Department of Atmospheric Science, University of Wyoming, Laramie, Wyoming, USA
[2]Argonne National Laboratory, Lemont, Illinois, USA
[3]Institute for Climate and Global Change Research, School of Atmospheric Sciences, Nanjing University, Nanjing,China
[4]International Center for Climate and Environment Sciences, Institute of Atmospheric Physics, Chinese Academy of Sciences, Beijing, China

*Correspondence to*: Xiaohong Liu (xliu6@uwyo.edu)

**Abstract.** A recent development in the representation of aerosols in climate models is the realization that some components of organic aerosol (OA), emitted from biomass and biofuel burning, can have a significant contribution to short-wave radiation absorption in the atmosphere. The absorbing fraction of OA is referred to as brown carbon (BrC). This study introduces one of the first implementations of BrC into the Community Atmosphere Model version 5 (CAM5), using a parameterization for BrC absorptivity described in Saleh et al. (2014). 9-year experiments are run (2003-2011) with prescribed emissions and sea surface temperatures to analyze the effect of BrC in the atmosphere. Model validation is conducted via model comparison to single-scatter albedo and aerosol optical depth from the Aerosol Robotic Network (AERONET). This comparison reveals a model underestimation of SSA in biomass burning regions for both default and BrC model runs, while a comparison between AERONET and model absorption Angstrom exponent shows a marked improvement with BrC implementation. Global annual average radiative effects are calculated due to aerosol-radiation interactions (REari; $0.13\pm0.01$ W m$^{-2}$) and aerosol-cloud interactions (REaci; $0.01\pm0.04$ W m$^{-2}$). REari is similar to other studies' estimations of BrC direct radiative effect, while REaci indicates a global reduction in low clouds due to the BrC semi-direct effect. The mechanisms for these physical changes are investigated and found to correspond with changes in global circulation patterns. Comparisons of BrC implementation approaches find that this implementation predicts a lower BrC REari in the Arctic regions than previous studies with CAM5. Implementation of BrC bleaching effect shows a significant reduction in REari ($0.06\pm0.008$ W m$^{-2}$). Also, variations in OA density can lead to differences in REari and REaci, indicating the importance of specifying this property when estimating the BrC radiative effects and when comparing similar studies.

**1 Introduction**

One of the key areas of uncertainty in climate models is their representation of aerosols (Anderson et al., 2003; Myhre et al., 2013). Aerosols tend to absorb (scatter) incoming solar radiation, which leads to warming (cooling) of the atmosphere. The directly emitted aerosol species (i.e. primary aerosols) are black carbon (BC), mineral dust, sea salt, and primary organic aerosol (POA or POM) – the latter of which broadly defines carbon-containing compounds that contain hydrogen and possibly oxygen (Bond et al., 2013). Aerosols that are products of chemical reactions (i.e. secondary aerosols) include sulfate, nitrate, ammonium, and secondary organic aerosols (SOA) (Kanakidou et al., 2005; Boucher et al., 2013). There are large uncertainties in aerosol observation and modelling which include the formation of SOA (Farina et al., 2010), aerosol and precursor gas emissions, aerosol aging, wet removal (Liu X. et al., 2012), and aerosol optical properties. In modeling studies, POA and SOA (more broadly organic aerosol, OA) are considered to be strongly scattering, but recent studies have shown that components of organic aerosols, known as brown carbon (BrC), can have strong absorption of UV and short visible light in the atmosphere. This absorption coupled with its large burden in the atmosphere (>3 times that of BC; Feng et al., 2013) indicate that BrC, in addition to dust and BC, is a significant absorber in the atmosphere.

While BC light absorption is assumed to vary weakly with changing wavelengths (Bergstrom et al., 2002), the absorption of BrC can be identified by its spectrally dependent absorption at shorter wavelengths (Kirchstetter et al., 2004). Early studies noted the presence of BrC in smoldering combustion (Patterson and McMahon, 1984), certain nitrated and aromatic aerosols (Jacobson, 1999), and low temperature, domestic coal combustion (Bond, 2001). Many studies have analyzed BrC in biomass burning laboratory experiments (Chen and Bond, 2010; Saleh et al., 2014; Pokhrel et al., 2016, Pokhrel et al., 2017), field studies in urban environments with biomass burning and urban aerosol sources (Kirchstetter et al., 2004; Liu S. et al., 2015), and aircraft measurements at different levels in the troposphere (Liu J. et al., 2014; Liu J. et al., 2015). In addition to primary BrC production, secondary BrC can be generated in aqueous-phase chemical reactions in clouds and from the photoxidation of volatile organic compounds (VOCs) (Limbeck et al., 2003; Ervens et al., 2011; Nakayama et al., 2013). There is also evidence that emissions from biomass burning (BB) and biofuel (BF) have a stronger BrC signal than emissions from diesel (Kirchstetter et al., 2004; Saleh et al., 2014). This may be due to the different burning processes of the two fuels, where the high pressure and high heat scenario of diesel combustion results in a more efficient breakdown of the fuel than the lower temperature, lower pressure burning of BB/BF; or it may be due to the different fuel types, where diesel consists of a mixture of smaller hydrocarbons and biomass consists of large polymers (Saleh et al., 2014).

While BrC production may change from fuel to fuel, its absorption strength can change even within a single fuel's emissions. The absorbing nature of BrC is correlated with the BC-to-OA ratio in BB emissions (Saleh et al., 2014; Lu et al., 2015), which corresponds to the burning conditions of the emission source. These studies show that higher BC-to-OA ratios correspond to stronger absorbing BrC, with higher BC-to-OA ratios indicating faster burning, hotter fires (e.g. savannah

fires) and lower BC-to-OA ratios indicating slower burning, smoldering fires (e.g. South America forest fires, boreal fires) (Akagi et al., 2011).

An added complexity in the study of aerosol radiative forcing (RF) is how to represent an aerosol's mixing state. Emitted primary aerosols that are often externally mixed (each particle consisting of one species) may become internally mixed aerosols (mixtures of species in each particle) as higher volatility compounds condense on and coat the surface of the primary aerosols within hours after emission (Reid et al., 1998). In three-dimensional global and regional aerosol models, this internal mixture can be treated as a well-mixed aerosol, where the optical properties of the aerosol are treated as a volume weighted mean of the optical properties of each internally mixed species (Ghan and Zaveri, 2007; Liu X. et al., 2016). Another treatment assumes a core-shell organization of aged aerosols where the primary aerosol core is coated by a volume mean internal mixture of higher volatility, stronger scattering secondary aerosols (Jacobson, 2001; Feng et al., 2013; Saleh et al., 2015). This can act to increase absorption given an absorbing center (e.g. BC) by refracting intercepted light into the primary core (Bond et al., 2006), also known as the lensing effect.

Lastly, BrC can undergo photochemical aging when oxidized by hydroxyl radical (OH) and when exposed to incoming solar radiation (Zhong and Jang, 2011; Lee et al., 2014; Zhong and Jang, 2014; Forrister et al., 2015; Zhao et al., 2015). This has the effect of an overall decrease in the absorption of BrC, with a half-life predicted at around 9-15 hours by in situ aircraft observations (Forrister et al., 2015), and 3 min to 5 hours from laboratory experiments (Zhong and Jiang, 2011; Lee et al., 2014; Zhao et al., 2015). Studies of BrC during the Green Ocean Amazon campaign during the Brazilian BB season observed a photochemical lifetime for BrC in sunlight of ~1 day (Wang et al., 2016). The initial aging of BrC can result in photo-enhancement or bleaching depending on the chemical composition of the aerosol, but with further photochemical aging the BrC absorption will decrease (Zhong and Jang, 2014; Zhao et al., 2015). This reduction is accelerated by higher relative humidities and is decelerated by the presence of nitrogen oxides ($NO_x$), the latter of which are a major emission from fossil fuel combustion and may play a role in the formation of absorbing SOA in smoke (Zhong and Jang, 2014).

The radiative effect of BrC has been estimated in a number of modeling studies using global chemical transport models combined with radiative transfer models (Feng et al., 2013; Lin et al., 2014; Wang X. et al., 2014; Saleh et al., 2015; Jo et al., 2016; Wang X. et al., 2018). In Feng et al. (2013), global mean absorption aerosol optical depth (AAOD) increases with two different BrC absorption assumptions: moderately (Chen and Bond, 2010) and strongly (Kirchstetter et al., 2004) absorbing, with 66% of BB/BF organic carbon emissions assumed to be BrC. In the case of strongly absorbing BrC, AAOD increases by 18% at 550 nm, and 56% at 350 nm. This study also shows that while BC contributes about 72% of the global atmospheric absorption, the strongly absorbing BrC could have a comparable effect (>20-50%) in regions dominated by biomass and biofuel burning, corresponding to a direct RF of about +0.11 W m$^{-2}$ due to strongly absorbing BrC (+0.04 W m$^{-2}$ for moderately absorbing BrC). A similar study by Lin et al. (2014) assumes that all SOA and POA is BrC, showing even larger contributions by BrC to aerosol absorption: BrC RF of +0.22 to +0.57 W m$^{-2}$ for the same moderate and strongly absorbing assumptions, respectively. These studies calculated RF by looking at the difference between top of the atmosphere

(TOA) shortwave flux, with and without BrC, over the time period spanning pre-industrial to present day. The higher RF in Lin et al. (2014) are likely due to the Lin et al. (2014) assumption of externally mixed aerosols as well as their consideration of SOA as absorbing (Saleh et al., 2015). In the modeling study by Saleh et al. (2015), BrC is incorporated by parameterizing OA refractive index (RI) based on the emitted BC-to-OA ratio from fuels representative of important BB regions (boreal

forests, grasslands, and croplands) (Saleh et al., 2014). This allows for spatial variation in BrC absorption as opposed to the globally constant moderate/strong absorption in Feng et al. (2013) and Lin et al. (2014). Use of this parameterization makes specifying the BrC fraction of OA unnecessary, as the BrC is dependent on the fuel source. Saleh et al. (2015) calculated the radiative effect due to aerosol-radiation interaction (REari), which represents the instantaneous effect of BrC on the Earth's energy balance and is calculated by comparing the energy balance at the TOA with and without BrC (Heald et al., 2014).

The BrC REari calculated in Saleh et al. (2015) varied between +0.12 W m$^{-2}$ and +0.22 W m$^{-2}$, with the variation attributed to core-shell and externally mixed assumptions, respectively. Recent work by Wang et al. (2018) included the Saleh et al. (2014) parameterization in addition to a BrC bleaching parameterization that ages BrC to 25% of its original absorption over ~1 day. This lower threshold reflects the observation that BrC doesn't fully bleach (Wang et al., 2016; Forrister et al., 2015). Including bleaching in the model resulted in a global REari of 0.05 W m$^{-2}$, significantly reducing the BrC radiative effect.

15       All these model representations show positive radiative effects in the atmosphere. However, by using chemical transport models coupled with radiative transfer models, these past studies neglect important BrC effects on clouds and surface albedo, as well as the more complex atmospheric dynamics found in Earth system models (Lin et al., 2014). Furthermore, only one of these studies includes the effect of BrC optical aging, which can greatly reduce the climate impact of BrC in these models. In this study, we incorporate BB and BF BrC into the Community Earth System Model (CESM) and

conduct an analysis of the radiative effect attributed to BrC over the time period 2003-2011. To simulate BrC we use the Saleh et al. (2014) parameterization to modify the BB and BF POA imaginary parts of the RI based on the BC-to-OA ratios in primary BB and BF emissions. The impacts of BrC on clouds through semi-direct effect are analyzed. We also implement a BrC bleaching parameter in the model similar to Wang et al. (2018) to analyze its effect on the BrC direct and semi-direct effects. It is important to mention that this study does not include absorption by BB SOA (Lin et al., 2014; Saleh et al., 2015)

or absorbing aromatic SOA (Wang X. et al., 2014; Jo et al., 2016; Wang X. et al., 2018). This is neglected due to the lack of aromatic SOA speciation in the model. As a result, the use of "OA" in regards to this study refers to primary organic aerosol. The paper is organized as follows: section 2 describes model and experimental setup as well as the modifications to the model; Section 3 introduces the model results of this study as a validation with the Aerosol Robotic Network (AERONET), radiative and climate effects due to BrC, and the sensitivity of the climate to BrC bleaching. Lastly, section 4 discusses the

results and presents conclusions from this study.

## 2 Model and Experiments

### 2.1 Model description

This study uses the Community Earth System Model (CESM), with the coupled Community Atmosphere Model version 5.4 (CAM5.4) (Hannay and Neale, 2015) and Community Land Model version 4 (CLM4) (Oleson et al., 2010). In Jiang et al. (2016) a similar model set up is used with the exception of its use of CAM5.3. CAM5.3 contains physics parameterization updates from the previous CAM versions, including a two-moment stratiform cloud microphysics scheme to predict mass and number mixing ratios of cloud liquid and cloud ice (Morrison and Gettelman, 2008), and a three-mode version of the Modal Aerosol Module (MAM3) to predict mass and number mixing ratios of aerosol components (Liu X. et al., 2012). Compared to the CAM5.3 version, some notable additions in the CAM5.4 version include a new prognostic precipitation scheme (Gettelman et al., 2015), new ice nucleation treatments (Wang Y. et al., 2014), and improved dust optical properties and emissions (Albani et al., 2014). This model also uses the 4-mode version of MAM (MAM4) (Liu X. et al., 2016). MAM4 consists of the following four lognormal modes (shown with their median size ranges and standard deviations): Aitken (0.015 – 0.053 μm, σ = 1.8), accumulation (0.058 – 0.27 μm, σ = 1.6), coarse (0.80 – 3.65 μm, σ = 1.8), and primary carbon (0.039 – 0.13 μm, σ = 1.6). The median sizes of aerosol modes are changed due to the microphysical processes (e.g., condensation and coagulation) while standard deviations for each mode are fixed. The OA from accumulation and primary carbon modes, which is used to represents BrC depending on its source (mentioned later), has a density in the model of 1 g cm$^{-3}$.

MAM4 differs from MAM3 in its inclusion of the primary carbon mode. This mode, which consists of the species BC and primary organic matter (POM), is emitted from the incomplete combustion of fossil and biomass fuels. In MAM4, this mode is transferred to the accumulation mode once it has aged. The primary carbon mode ages by acting as a nuclei for the condensation of sulfuric acid ($H_2SO_4$) vapor, ammonia ($NH_3$), and semi-volatile organics, as well as through collision and coalesce with Aitken and accumulation modes. This acts to both increase the size of the aerosol as well as increase the hygroscopicity of hydrophobic aerosols such as BC. The optical calculations in the model consider these aged particles to be internal mixtures of aerosols within the mode, and the RI of the accumulation mode, as well as the other 3 modes, is calculated as a volume-weighted mean of the refractive indices of all of the aerosol's components within the mode (Liu X. et al., 2012; Liu X. et al., 2016).

## 2.2 Model Modifications

### 2.2.1 Source separation

Code modifications were made to allow implementation of the new POM RI for BB and BF emissions. This involved separating the emissions of BC and POM into three sources: biomass burning (BB), fossil fuels (FF), and biofuels (BF). In the model, BB corresponds to the Global Fire Emissions Database version 3.1 (GFED 3.1; Giglio et al. (2013)) fire emissions; FF corresponds to energy, industry, ship, and transport emissions from IPCC AR5; and BF corresponds to agricultural, domestic, and waste management emissions from IPCC AR5. This modification was based on a similar separation of emissions implemented in CAM5.3 (Jiang et al., 2016). Figure 1 shows the contributing regions and burdens of

POM from BB, BF, and FF sources. The GFED 3.1 emissions were used in this study to allow for direct comparison between this study and Jiang et al. (2016). The more recent GFED 4 emission dataset shows an 11% global increase in fire emissions from GFED 3.1 (Werf et al., 2017), which may result in a slightly stronger climate impact from biomass burning aerosols than that shown in this study.

### 2.2.2 BrC refractive index

Another modification involved the inclusion of a parameterization for BrC in CAM5.4. The parameterization is described in Saleh et al. (2014) and determines the imaginary refractive index (RI) of POM based on the BC-to-OA ratio of the BB and

BF emissions. A later experiment by Lu et al. (2015) derives a similar parameterization based on many different BB datasets, including that of Saleh et al. (2014). The Saleh et al. (2014) RI is calculated based on the following equations

$$RI = 1.7(\pm 0.2) + k_{OA}i = 1.7(\pm 0.2) + k_{OA,550}\left(\frac{550}{\lambda}\right)^{w} i,$$

(1)

$$k_{OA,550} = 0.016 \bullet \log_{10}(BCtoOA) + 0.04,$$

(2)

$$w = \frac{0.21}{(BCtoOA + 0.7)},$$

(3)

where Eq. (1) describe the real (1.7) and imaginary ($k_{OA}$) parts of the RI. Equation (2) represents the imaginary RI at 550 nm ($k_{OA,550}$), and Eq. (3) represents the wavelength dependence of $k_{OA}$ ($w$). The parameterization describes decreasing wavelength dependence and increasing BrC absorption as the emission BC-to-OA ratio increases. This relationship was derived for the wavelengths of 370-950 nm and a laboratory derived BC-to-OA ratio range of $0.01 - 0.5$. In the model, the parameterization is applied over this BC-to-OA range and for the CAM waveband midpoints of 304-1010 nm (wavebands 8-

12 in the CAM RRTMG_SW model (Neale et al., 2012)). Uncertainty in $k_{OA}$ from this parameterization is associated with the lab measurements of the particle mass, the range in assumed complex refractive index for BC, the mixing state of BC and OA, the measured real part of the OA refractive index, and the measured absorption coefficients used in optical closure calculations (Saleh et al., 2014).

The parameterization in CAM only changes the RI of POM (emissions of POM and density (1 g cm$^{-3}$) remain

unchanged) and was applied in the part of CAM5.4 that calculates modal aerosol optical properties using a parameterization described in Ghan and Zaveri (2007). This parameterization calculates the specific scattering, specific absorption, and asymmetry parameter of aerosols in a mode as a general internal mixture of components, bilinearly interpreting these optical

properties from offline Mie calculations. These calculations are a function of the wet surface mode radius as well as the wet refractive index of each mode. The code modification to implement a changing RI for BrC is implemented before the bilinear interpretation. When the model calls the RI for BB or BF POM from an input physical properties (phys_prop) file, the parameterization changes the imaginary part of the RI at the wavelengths 304-1010 nm based on the calculated BC-to-OA ratio at each time-step for each grid point and level. The BC-to-OA ratio is calculated based on each source's ratio (BB or BF) to preserve the initial emission ratios and minimize error due to inter-source mixing (see Fig. 2 for BB and BF BC-to-OA ratios). As shown in Fig. 2, BB BC-to-OA ratio is uniformly distributed in the range of 0.07 to 0.095 with a global mean of 0.084. BF BC-to-OA ratio is much higher, varying in the range of 0.1-0.3 with a global mean of 0.185.

The BRC_CNST experiment was run with the same Saleh et al. (2014) RI parameterization, but the BC-to-OA ratio was set to 0.08 (approximate global column average biomass burning BC-to-OA) at each time step, grid point, and level. This results in a constant imaginary RI for BB and BF POM.

A few assumptions in this model simulation introduce uncertainty in the representation of BrC in CESM. One of those assumptions is neglecting absorption by BB SOA (Lin et al., 2014; Saleh et al., 2015) or absorbing aromatic SOA (Wang X. et al., 2014; Jo et al., 2016; Wang X. et al., 2018), which is neglected due to the lack of SOA speciation in the model. This assumption, in conjunction with the use of GFED 3.1 instead of GFED 4, may act to underestimate the climate effect due to BrC. Another assumption is the model use of a volume mixing assumption, which may act to overestimate aerosol light absorption (Jacobson, 2000; Adachi et al., 2011). We also assume that the BC-to-OA ratio in transported smoke is similar to BC-to-OA from the source region, allowing for the use of a BC-to-OA ratio at each gridcell at every time step to calculate $k_{OA}$ in each gridcell. The uncertainty in $k_{OA}$ associated with this assumption is small (<10% for BB emissions assuming transport from the Equator to the Arctic (not shown)) and is assumed to be negligible.

Another source of uncertainty when considering an absorbing aerosol in the model is the aerosol's vertical distribution. CAM5.4 uses six vertical injection heights for wildfire emissions described in Detener et al. (2006): 0-100 m, 500-1000 m, 1-2 km, 2-3 km, and 3-6 km. These fire emission heights depend on the geographic location of the fire and the vegetation type derived from GFED, with the highest plumes corresponding to boreal fires. If BrC is lofted over a more reflective surface such as a cloud, its shortwave radiative forcing will be more positive than if it stays below the cloud or remains lower in the atmosphere. A counterbalancing effect is a more negative longwave forcing at higher levels in the atmosphere (Penner et al., 2003). The vertically sensitive semi-direct effects of BrC (i.e., changes in atmospheric stability and cloud cover due to atmospheric heating by BrC) are discussed in more detail in section 3.2. Comparisons between the total OA (POA + SOA) vertical distribution and aircraft observations in Shrivastava et al. (2015) show that the standard CAM5 aerosol treatment largely underestimates Arctic biomass OA, possibly due to the model neglecting important SOA contributions from biomass burning. This could lead to an underestimation of BrC radiative effects due to lower BrC concentrations at all levels of the model. Vertical profiles of aerosols, cloud fraction, and heating rates in the model are shown over 6 regions with strong BrC radiative effect due to aerosol radiation interaction (Fig. S2).

### 2.2.3 BrC Bleaching Parameterization

The BrC bleaching parameterization is based on that applied by Wang et al. (2018). The half-life of BrC is assumed to be
about 1 day with the aging of BrC dependent on the atmospheric concentration of OH ([OH], molec cm$^{-3}$). Equation (4) describes this aging process in the model:

$$k_{OA,t+\Delta t} = k_{OA,t} \bullet \exp\left(-\frac{[OH] \bullet \Delta t}{5 \bullet 10^5}\right),$$

(4)

where $k_{OA,t+\Delta t}$ and $k_{OA,t}$ are the imaginary part of the RI of BrC at times $t$ and $t+\Delta t$, and $5\bullet10^5$ represents the typical daytime
[OH] (Wang et al., 2016). This parameterization was designed to match observed lifetimes of BrC in the two field campaigns SEAC$^4$RS (Forrister et al., 2015) and GoAmazon2014/15 (Wang et al., 2016). While the parameterization depends on OH concentration in the atmosphere, by matching the BrC lifetime to observations the parameterization also includes photochemical oxidation and other bleaching effects that may have been active in the observed smoke plumes. This is true of the regions in which the observations were taken, but may not hold true for global sites or seasons with lower insolation.
Uncertainty in this parameterization is associated with the low availability of observational data, and could be improved with more field measurements of BB smoke aging at different latitudes.

### 2.3 Experimental design

The model was run at a horizontal resolution of 0.9˚ latitude by 1.25˚ longitude with 30 vertical levels. The simulation period covers 9 years from 2003 to 2011 with prescribed monthly sea surface temperature and sea ice. The emissions consist of daily BC, POM, and sulfur dioxide (SO$_2$) emissions from GFED 3.1 (Giglio et al., 2013), vertical distribution of fire emissions based on the AeroCom protocol (Detener et al., 2006), and anthropogenic aerosol and precursor gas emissions from IPCC AR5 (Lamarque et al., 2010). The model runs consist of a spin-up year (2003), followed by a 9-year run (2003-
2011) using a concatenated version of the emissions files for each species and emission type. Four experiments, with 5 ensemble members each, were run following the aforementioned setup. Ensemble members are varied by applying different initial temperature perturbations of the order $10^{-14}$. NOBRC is the control experiment with the default model configuration, while BRC includes a parameterization for BrC that takes into account a varying, BC-to-OA ratio dependent RI. BRC_CNST includes a parameterization for BrC that assumes a constant imaginary RI based on the approximate global
average BC-to-OA ratio of 0.08, derived from biomass burning emissions. BRC_BL includes the BrC parameterization as well as a BrC bleaching parameterization. Table 1 describes the different model runs.

### 2.4 Radiative effect calculation

The direct radiative effect and indirect effect of BrC are calculated using the method recommended in Ghan (2013). This method breaks the total radiative effect into changes due to aerosol-radiation interactions (REari), aerosol-cloud interactions (REaci), and surface-albedo changes (REsac) (note: REsac is not analyzed in this paper because the model does not take into account snow albedo reduction via BrC deposition on snow).

$$\Delta F = \Delta(F - F_{clean}) + \Delta(F_{clean} - F_{clear,clean}) + \Delta(F_{clear,clean}),$$
$$\quad\quad\text{(REari)}\quad\quad\quad\quad\text{(REaci)}\quad\quad\quad\quad\text{(REsac)}$$
$$(5)$$

In Eq. (5), $\Delta$ represents the difference between the model runs with and the model run withough BrC, in this case represented by BRC (BRC_CNST, BRC_BL) – NOBRC. The differences in each of the parentheses are conducted for each model run represented in $\Delta$. The variable F represents all-sky, top of the atmosphere (TOA) radiative flux (longwave + shortwave), $F_{clean}$ is the same but without aerosols, calculated from the diagnostic radiation call in the model run, and $F_{clean,clear}$ is removing both aerosols and clouds.

15        While this method calculates aerosol indirect effects – changes in cloud microphysics due to more/less numerous cloud condensation nuclei – and aerosol semi-direct effects – changes in cloud environments due to radiation interaction of the aerosols – (collectively REaci), when it is applied to the experiment with absorbing versus non-absorbing organics, REaci represents only the semi-direct effect of BrC (indirect effects of organics are already accounted for in these two experiments). This effect describes the ability of absorbing aerosols to change the lifetimes of clouds by changing the static

stability of the troposphere and limiting incoming solar radiation to the surface; also, these aerosols may help to evaporate cloud droplets when heating the cloud levels (Ackerman et al., 2000; Koch and De Genio, 2010).

## 3 Model Results

### 3.1 Model validation

Four model simulations (i.e. NOBRC, BRC, BRC_CNST, BRC_BL) are compared to level 2.0 Aerosol Robotic Network (AERONET, http://aeronet.gsfc.nasa.gov) observations from nine different sites, located in three different BB regions: North American and Asian boreal forests, South America, and Africa. The AERONET data is averaged over the period 2003-2011

to match the model simulation period. The model directly outputs the 440 nm single scattering albedo (SSA), aerosol optical depth (AOD), and absorption aerosol optical depth (AAOD) used in this analysis.

        The relationship between AERONET and SSA (Fig. 3) and AOD (Fig. S3) from the CAM5.4 NOBRC experiment is the same as in the Jiang et al. (2016) comparison with CAM5.3. As with Jiang et al. (2016), CAM5.4 does a reasonably good job of simulating AOD and SSA, although there are some regions of disagreement, possibly due to incorrect fire

emissions, excessive aerosol scavenging by liquid-phase clouds, or inaccurate representation of aerosol optical properties and/or size (Jiang et al., 2016; Yu et al., 2016).

Another possibility for the disagreement between model and AERONET may be due to error associated with AERONET. The AERONET cloud screening process may introduce error by screening aerosol plumes that have highly variable optical properties (Smirnov et al., 2000). Furthermore, the inversion algorithm used to calculate SSA from AOD (Dubovik and King, 2000) introduces error in retrieving BB SSA in low aerosol concentration situations (Dubovik et al., 2000). Also, the dynamics in the model are going to be different than those influencing the AERONET sites, which could lead to a difference in aerosol exposure at these locations.

When comparing the four model experiments to AERONET AOD **(**Fig. S3)**,** there is good agreement between the model with and without BrC. This is due to the fact that model AOD is dominated by aerosol scattering, and any changes in absorption with the incorporation of BrC are small by comparison. Overall there is little obvious effect on AOD due to BrC. A much stronger effect can be seen when looking at AAOD, with peak BrC effects in bomass burning seasons (Fig. S4).

When looking at SSA (Fig. 3) the differences between the four model runs are more apparent. In all of the regions (Arctic, Africa, South America), incorporation of BrC (BRC, BRC_CNST, BRC_BL) decreases SSA with respect to the default CAM5.4 model (NOBRC). This is especially true of biomass burning seasons. One would expect this to be true given the large burdens of OA released at these times coupled with the modification to their RI. This seasonality is less apparent at Ascension Island, possibly due to less concentrated OA from diffusion and scavenging during transport to the island, leading to a smaller effect from BrC.

The bleaching effect in BRC_BL reduces its absorption during these seasonal peaks, placing it closer to the NOBRC simulation than either BRC or BRC_CNST. When looking at the African and South American sites, especially those further away from the BB source regions (Fig. 3b-c), the BRC_BL is more similar to NOBRC; in the Arctic, BRC_BL tends to be closer to the BRC simulation. This can be explained by the higher [OH] in the tropics resulting in a faster bleaching of BrC than in the Arctic. Table S1 shows [OH] in different regions and the half-life of BrC due to the bleaching effect in these regions, which ranges from 0.37 days (southeast Asia) to 2.09 days (Arctic).

Another effect that can be seen in the SSA plots is a subtly lower SSA (i.e. stronger absorption) in the Arctic regions (Fig. 3g-i) for BRC_CNST than for BRC. This may relate to the constant BC-to-OA of 0.08 in BRC_CNST, which is higher than the column burden average BC-to-OA at these high latitudes (Fig. 2). A higher BC-to-OA relates to stronger absorption based on the Saleh et al. (2014) parameterization. While the differences between BRC and BRC_CNST are quite small, this presents the possibility of overestimation of the BrC effect at Northern high latitudes with BRC_CNST.

While the SSA comparison does not show improvement with BrC incorporation, agreement between model and AERONET absorption Angstrom exponent (AAE) is better with BrC (Fig. 4). AAE is calculated based on the two wavelegths 440 nm and 675 nm ($\lambda_1$ and $\lambda_2$, respectively) and the measured absorption coefficients at the two different wavelengths ($b_{abs}(\lambda_1)$ and $b_{abs}(\lambda_2)$).

$$AAE = \frac{-\ln(b_{abs}(\lambda_1)/b_{abs}(\lambda_2))}{\ln(\lambda_1/\lambda_2)}$$

,

(6)

It describes the wavelength dependence of the measured emissions, with AAEs greater than one representing higher wavelength dependence (i.e. larger differences in $b_{abs}(\lambda_1)$ and $b_{abs}(\lambda_2)$). As a result, higher AAEs can be used to identify the presence of BrC (Bond, 2001; Kirchstetter et al., 2004; Chen and Bond, 2010; Bond et al., 2013).

    Default CAM (NOBRC) tends to overestimate the AAE of the previously mentioned biomass burning sites when AERONET predicts low values; and vice versa (Fig. 4a). In the BRC, BRC_CNST, and BRC_BL model runs, the

underestimation of AAE by CAM is improved, especially in the South American and Arctic regions (Fig. 4b-d). This is consistent with Feng et al. (2013) who showed that including different values for absorbing OA drives the model AAE closer to that of AERONET.

    The large African outliers in this comparison are all from the Ascension Island site (Fig. 4e-h). Comparison of AAE and scattering Angstrom exponent (SAE) for the AERONET sites (Fig. S5) suggest that the Ascension Island aerosols are a

large particle/low absorption mixture (Fig. 8 in Cappa et al., 2016). When the same comparison is made for the BRC model run (Fig. S6), the Ascension Island aerosols are identified by the Cappa et al. (2016) classification as a large particle/BC mixture. The large particle trend at Ascension Island can likely be attributed to sea salt aerosol, while the difference between the model and AERONET may be due to more smoke transport to the island in the model. Two other possibilities for the disagreement were previously mentioned: error introduced to AERONET due to cloud screening processes and model bias in

the circulation patterns.

    Other comparisons between the model and observational AAE are shown in Table 2. These observations are from Ascension Island (Zuidema et al., 2018), a southwestern US (New Mexico) wildfire (Liu S. et al., 2014), and continental US BB and prescribed burns from the SEAC[4]RS campaign (Mason et al., 2018). These comparisons are largely qualitative given the coarse resolution of the model (~100 km) compared to the much finer wildfire spatial resolution. Both the Ascension

Island and New Mexico observations are from the surface, while the continental US observations are made via aircraft. The Ascension Island and SEAC[4]RS US BB/BF observations were made during different years than represented in the model run, so our comparison operates under the assumption that these regions have consistent seasonal smoke exposure from 2003-2016. The bleaching effect (BRC_BL) drives the model closer to the observations over Ascension Island, but still acts to overestimate the AAE along with BRC and BRC_CNST. This could suggest either too much BrC transport to this region

in the model, too weak a bleaching effect, or too much of some other aerosol with a higher wavelength dependence (i.e.,

dust). The models underestimate AAE over the US, especially in the BRC_BL and NOBRC model runs. This could be due to an influence of fossil fuel emissions in the model grid cell driving the AAE closer to 1.

Other observations of forest fires from Los Alamos National Laboratory (LANL) in New Mexico from 2018 show the variability of measured fire AAE. These include the Buzzard Fire, the Ute Fire, and the San Antonio Fire (with average
AAEs of 1.44, 1.42, 2.38, respectively) (Romonosky et al., submitted for publication). Due to their being outside the model run period and their intermittent nature, these fires are not directly compared to the model. However the Las Conchas model comparison from Table 2 is compared to the other fires assuming a similar regional fuel composition in the model. The model with the bleaching effect is closer to the Ute and Buzzard fires than to the Las Conchas and San Antonio fires, and overall the inclusion of BrC brings model AAEs closer to the New Mexico fire measurements.

### 3.2 BrC radiative effect

One of the effects of adding BrC to the model is an increase in atmospheric absorption due to BB/BF emissions. This can be
seen in Fig. 5a which shows the difference in AAOD between the BRC and NOBRC model runs. The max AAOD over southern Africa is about 0.024, which makes up approximately 1/3 of the total AAOD in this same region (Fig. S7a). As with Fig. 4, Fig. 5b shows a global increase in AAE due to BrC. These effects are the strongest over the southern African BB region and in the Arctic, with the strongest AAE increases over the Arctic (Fig. 5b) correlated with weaker AAOD in Fig. 5a. Vertical cross-sections of aerosol absorption coefficient and AAE changes due to BrC (Fig. 6), show the vertical extent
of BrC. Zonal BrC absorption is dominated by the African and South American biomass burning regions, with visible aerosol transport to the Arctic from boreal fires (Fig. 6a). While absorption over the Antarctic is nearly zero, upper level transport of dilute BrC to the south can be inferred from the AAE changes in Fig. 6b.

Fig. 7a shows REari of BrC from the BRC model run. The global annual average of REari is $0.13 \pm 0.01$ W m$^{-2}$ ($\pm 1$ $\sigma$ standard deviation), which indicates a global warming effect due to BrC. The maximum forcing (~1.75 W m$^{-2}$) occurs off
the west coast of southern Africa where lofted BB emissions are transported over low-lying semi-permanent stratocumulus cloud deck off the coast of Namibia and Angola. This maximum forcing can be explained by an enhancement of BrC absorption as incoming solar radiation is not only absorbed at the top of the smoke plumes, but is also reflected off the clouds and is absorbed by the smoke plumes as outgoing radiation from the cloud top (Abel et al., 2005; Zhang et al., 2016). Significant absorption also occurs over South America BB regions (~0.3 W m$^{-2}$), southeastern Asia BF sources (~0.5 W m$^{-2}$),
and regions over the Arctic Ocean and Greenland (0.1-0.3 W m$^{-2}$).

While the effects of REari correspond well to emission sources of absorbing aerosol, REaci presents a more complicated picture to diagnose (Fig. 7b). There is a weakly positive REaci global average ($0.014 \pm 0.04$ W m$^{-2}$) suggesting that the BrC absorption has a negative impact on low cloud formation and lifetimes in the model, reducing the cooling effect of clouds globally. Some regions where the REaci is positive include South America (~2 W m$^{-2}$), the Gulf of Mexico (~2.5

W m[-2]), Kazakhstan (~1.5 W m[-2]), parts of southwest Alaska (~1.5 W m[-2]), much of Australia (~2 W m[-2]), and the Weddell Sea (~2 W m[-2]). This positive effect is almost balanced by the negative REaci (up to -2 W m[-2]) over the ocean (e.g., northeastern Pacific, Indian Ocean, and Antarctic Coast) and in parts of South America and northeastern China. A negative REaci suggests an increase in cloud formation as a result of the BrC addition to the model. We note that most of these REaci are not statistically significant.

Johnson et al. (2004) used high-resolution large eddy simulating (LES) model experiments of absorbing aerosols in and above the boundary layer to show that REaci could be positive if absorbing aerosols within the boundary layer heated the air, evaporating cloud liquid water and cutting off the surface moisture source via a decoupling of the boundary layer. They also found that REaci could be negative by enhancement of an elevated stable layer due to absorbing aerosols above the boundary layer. By creating a strong inversion and increasing the potential temperature above cloud top, the presence of elevated absorbing aerosols allowed for more persistent low-level clouds by creating less favorable conditions for cloud top entrainment. Another study by Sakaeda et al. (2011) looked at the semi-direct effect of BB aerosols over Africa and found that REaci was typically negative over the ocean and positive over land. In their study they found that the low-level marine cloud cover is increased as a result of increasing low-level tropospheric stability, caused by the direct radiative heating above clouds and the resulting cooling of the surface. This increases the relative humidity in the lower troposphere. Over land they found positive REaci to indicate an increased static stability due to warm aerosols aloft, which reduces convection and slows development of convective precipitation. Possible mechanisms for the significant regional REaci will be addressed in section 3.4.

Interestingly, semi-direct effects are absent from the southern African region where BrC absorption dominates. Within REaci, indirect effects such as cloud droplet effective radius reduction due to enhanced concentrations BB aerosols have greater effects than semi-direct effects (Jiang et al., 2016; Lu et al., 2018). With this in mind, the lack of semi-direct effects over Africa could be due to the dominating indirect aerosol effects in this region from OA, which may be acting to mask the smaller semi-direct effects of BrC. Another possibility is that the absorption of BC (~4 W m[-2]; Jiang et al., 2016), being stronger in this region than the absorption due to BrC (~1.75 W m[-2]), is the most important factor in semi-direct effects. Both of these hypotheses may lead to fewer occurrences of significant cloud change due to BrC.

### 3.3 BrC effects on SW radiation, surface temperature, clouds and precipitation

Looking at differences between BRC and NOBRC for specific climate variables helps clarify possible mechanisms for the previously discussed radiative effects. In Fig. 8a, the distribution of atmospheric absorption is largely synonymous with REari with a global mean of $0.29 \pm 0.01$ W m[-2]. The atmospheric absorption is most apparent over the south Africa BB and S. E. Atlantic outflow regions (~3 W m[-2]), and over BF emission sources on the edge of the Tibetan plateau (~1.5 W m[-2]). Surface SW flux reduction due to BrC absorption can be seen in the S. E. Atlantic outflow region and the African BB site (Fig. 8b). Also, surface SW flux change (Fig. 8b) corresponds well to the REaci, indicating an increase (decrease) in solar

radiation reaching the surface with positive (negative) REaci. But what causes changes in REaci? To understand this, it is important to look at the driving forces behind these radiative effects: clouds and precipitation.

The vertically integrated low-level cloud fraction for clouds lower than 700 mb (LCF; Fig. 8c) shows an inverse correlation with REaci in some regions, indicating that a decrease in LCF may be correlated partially with an increase in
REaci, or vice versa. This is most apparent in South America, western Australia, the Middle East, and northeastern China, with smaller pockets of significant change in Kazakhstan, northern Canada, and Antarctica. Mid-level (between 700 and 400 mb) cloud fraction (MCF; Fig. 8d) is consistent with LCF in the Arctic, parts of South America, western Australia, northeastern China, and Kazakhstan. Some of the regions where it differs from LCF are off the northwest Coast of the United States, in northern Russia, and off the southeast tip of Australia.

Globally there is little change in large-scale precipitation (PRECL; Fig. 8e)  (0.0003±0.001 mm). Regionally, there are significant changes in PRECL corresponding to changes in LCF. Examples include the Arctic Ocean (-0.15 to 0.1 mm day$^{-1}$), Iran (0.15 mm day$^{-1}$), off the East Coast of the US (0.2 mm day$^{-1}$), in parts of South America (0.2 mm day$^{-1}$), near Kazakhstan (-0.15 mm day$^{-1}$), and off the West Coast of the US (-0.4 mm day$^{-1}$). Increases in PRECL in the Arctic may indicate increased snowfall due to colder temperatures at these latitudes. There is a significant decrease in PRECL off the
West Coast of the US, and while this doesn't correspond to low-cloud fraction, it does agree with a decrease in MCF (Fig. 8d).

Change in precipitation from convective clouds (PRECC; Fig. 8f) has a different distribution than PRECL. Significant regions of change are found in the tropics with large reductions in PRECC found along the Gulf Coast of Mexico (-0.4 mm day$^{-1}$), in western Australia (-0.2 mm day$^{-1}$), and in parts of South America (-0.1 mm day$^{-1}$). There are also
significant changes following the northern Atlantic storm track (-0.3 mm day$^{-1}$) and off the West Coast of the US (-0.2 mm day$^{-1}$). While there are cells of significant PRECC increase in parts of Bangladesh (0.2 mm day$^{-1}$), China (0.1 mm day$^{-1}$), and Africa (0.1 mm day$^{-1}$), the global mean of -0.007±0.0015 mm day$^{-1}$ indicates a slight decrease in PRECC with the addition of BrC.

Changes in liquid water path (LWP; Fig. 8g) are consistent with changes in LCF and PRECC. The most negative
LWP changes are over the land in South America (-4.5 g m$^{-2}$), Australia (-3.5 g m$^{-2}$), and the Gulf of Mexico (-5 g m$^{-2}$), corresponding to reductions in PRECC (Fig. 8f). This is consistent with Sakaeda et al. (2011), who correlated large reductions in convection with increased static stability resulting from elevated BB aerosols. Significant increases in surface air temperature (TS; Fig. 8h) appear in South America (0.4 ˚K), the Gulf of Mexico (0.4 ˚K), and Austrailia (0.4 ˚K). These regions correspond to decreasing PRECC, PRECL, and LCF, indicating surface warming in regions with decreasing cloud
cover and precipitation. Significant increases in TS over Alaska (0.5 ˚K) and northern Africa (0.25) aren't correlated with cloud cover change and may be due to surface changes due to changing dynamics in the model with BrC incorporation. Similarly, TS significantly decreases over the Rocky Mountains and the northern Plains of the US (-0.6 ˚K), but does not seem to correspond to significant changes in cloud cover or precipitation. This change is unclear but may indicate a change in surface albedo in the mountainous regions, possibly due to increased snowfall.

It is important to note that the model output does not give a complete picture of temperature or precipitation changes due to the prescription of sea surface temperatures. It would be worth revisiting this study with a fully coupled atmosphere and ocean model to allow BrC absorption to affect the air-sea interactions in the model.

**3.4 Vertical changes due to BrC semi-direct effects**

To understand how the BRC model BrC plays a role in aerosol-cloud interactions we looked at vertical profiles of land (western Gulf of Mexico (GM), South America (SA), northeastern China (NEC)) and oceanic regions (Weddell Sea (WS), western Antarctic Coast (AC), northeast Pacific (NEP)) with significant positive and negative BrC REaci (Fig. 9). The

vertical profiles come from the averaged grids in these regions with greater than a 0.9 confidence level (Fig. 10 and 11).

In the Antarctic oceanic regions, there is little influence from BB and BF POA (i.e., sources of BrC) (Fig. 10a) and there is a positive vertical motion associated with the Ferrel Cell convergence zone (Fig. 10e). The NEP site has a larger aerosol influence as well as negative vertical motion due to the descending branch of the Hadley circulation. WS low-level cloud fraction (Fig. 10b) and potential temperature (Fig. 10c) correspond with the greater annual average ice fraction in the

WS (63% versus 15% off the AC). The heating rate in all three regions corresponds to the local cloud fraction maxima, indicating strong contribution from latent heat release (Fig. 10d). In all of the regions, the upper level concentration of BB and BF POA increases by about 10-15% (Fig. 10f) indicating a change in atmospheric circulation with BrC implementation. This is backed up by positive changes in omega (~20% max) (Fig. 10j) and decreases of ~4% (increases of ~2%) in high (low) level clouds (Fig. 10g). The NEP region shows a decrease in aerosol concentration at lower levels (5%), possibly

indicating a strengthening of the Hadley cell down welling, which may change advection of aerosol into the region or inhibit advection due to the high-pressure enhancement. In the WS there is a decrease in low-level clouds (3% max) and an increase in upper level clouds (1%), and off the AC the low level clouds are enhanced (1%) (Fig. 10g). Cloud fraction changes at all three sites can be related to changes induced in potential temperature (Fig. 10h) and heating rate (Fig. 10i). Due to low concentrations of aerosol in these regions the main driving force behind the significant REaci may be due to changes in

vertical motion (Fig. 10j).

Increased cloud cover near the surface may lead to the negative REaci off the AC, which is consistent with strengthening upward velocities and can also be seen by a slight significant increase in LCF (Fig. 8c). The factors leading to a positive REaci in the WS may be the increased high cloud cover as well as the decreased low cloud cover correlating with descending air near the surface (Fig. 10j). Why the air descends in this region is unclear, but may be related to the larger ice

fraction in the region contributing to an inversion that may decouple the lower levels from the upper level circulation changes as observed in a WS field campaign by Andreas et al. (2000).

The land sites that are selected have a much stronger influence from BB and BF POA than the Antarctic Ocean sites (Fig. 11a). NEC is marked by slightly higher midlevel cloud fractions and lower temperatures than the GM or SA (Fig. 11b-c), and is in a region of descending air aloft (Fig. 11e). Both the GM and SA are located in latitudes near the descending

branch of the Hadley circulation but are marked by different regional circulations: SA has an annual average upwelling (Fig.

11e, 12a) influenced by lee-side cyclogenesis (Mendes et al., 2007) while GM has annual average downwelling influenced by the Hadley circulation (Fig. 11e, 12a). Figure 11f shows changes in POA concentrations ranging from ~4% decrease to ~1% increase near the surface. Regions with positive REaci have decreased cloud fraction (~7% max) while regions with negative REaci have increased cloud fraction (~5% max) (Fig. 11g). In NEC the increased cloud fraction corresponds to cooler surface temperatures (Fig. 11h) and an increased heating rate (Fig. 11i; associated with latent heat release). In SA and GM, decreasing clouds at lower levels correspond to an increased heating rate (Fig. 11i), potentially due to the presence of absorbing BrC in the atmosphere overpowering the reduction in latent heating rate. Decreasing upper level clouds result in a cooling of the upper atmosphere, which may be attributed to a weakening of the high cloud atmospheric warming effect. Cloud fraction increase (decrease) corresponds to negative (positive) changes in omega in the selected regions (Fig. 11j). A strong enhancement of vertical velocity at the surface in the GM (Fig. 11j) may be attributed to the influx of BrC into the region below cloud level (Fig. 11f) and the high heating rate near the surface (Fig. 11i).

Figure 12b shows the change in 500 hPa omega ($\omega$) with the incorporation of BrC. There is little significant change in $\omega$ over the regions selected here with the exception of AC, SA, and NEP, where incorporation of BrC acts to enhance both sinking (SA and NEP) and rising motion (AC). This may be due in part to an enhancement of the global rising and sinking motions due to enhancement of convection by absorbing aerosol (Perlwitz and Miller, 2010). Of note is the large swath of sinking motion off the West Coast of North America, which correlates to changes in MCF, PRECL, and PRECC in Fig. 8. The NEP cooling region corresponds to the significant increases in LCF (8c) and LWP (8g) within this region of increased $\omega$.

**3.5 Comparison of BRC, BRC_CNST, and BRC_BL climate effects**

There are two methods of BrC implementation that have been used in CAM5. Other than the code modification described in section 2.2.2 (BRC model run), the other option has been to use offline calculated phys_prop files with a constant OA RI that is calculated based on a constant BC-to-OA ratio of 0.08 (BRC_CNST model run; Yan Feng, personal communication). The assumption of a constant BC-to-OA ratio may be inaccurate in regions that have a BC-to-OA ratio higher or lower than 0.08. To see if this results in a significant change between the different model simulations, the global climate effects of both methods are examined to determine how they differ. We also show the effect of bleaching (BRC_BL) on these regions but do not discuss it in this section.

Table 3 shows the average of different components of BrC RE for the tropics (25˚S to 25˚N), the Arctic (60˚N to 90˚N), and the entire globe. REari is similar between the two methods in the tropics (~0.14±0.01 W m$^{-2}$ as well as globally (0.13±0.01 W m$^{-2}$). In the Arctic, REari for BRC_CNST (0.23±0.02 W m$^{-2}$) is slightly larger than REari for BRC (0.21±0.04 W m$^{-2}$). This difference in the Arctic may be due to a larger BC-to-OA ratio in BRC_CNST when 0.08 is assumed. In the model, BB BC-to-OA ratio ranges from <0.07 to 0.072 (Fig. 2a) in the Arctic, which would lead to weaker absorption in BRC than BRC_CNST. This may also explain the similarity between REari in the tropics where model BB BC-to-OA ratio

is similar to 0.08 with a range of 0.076 to 0.096 (Fig. 2a). While BF undoubtedly plays a role in these regions, the effect is likely less important in large global sectors than BB due to its lower emissions.

Looking at the global and Arctic regions in Table 3, the REaci associated with BRC_CNST is lower than that associated with BRC. This comes with the caveat that both have very large standard deviations, lowering the significance of this comparison.

Table 4 shows the global averages of selected variables associated with atmospheric SW transmittance, clouds, precipitation, and surface temperature. Like REari, global atmospheric absorption is very similar between BRC (0.29±0.01 W m$^{-2}$) and BRC_CNST (0.28±0.03 W m$^{-2}$). The largest difference between the two model runs seems to be related to differences in aerosol-cloud interactions. In Table 4, LWP change is higher in the BRC_CNST (0.06±0.09 g m$^{-2}$) than in BRC (0.004±0.1 g m$^{-2}$) model run which seems to correspond to higher LCF (0.06±0.03 %) and MCF (-0.006±0.02 %) changes in BRC_CNST. Consistent with increased cloud cover and LWP is a lower surface flux in BRC_CNST (-0.18±0.06 W m$^{-2}$). While cloud cover and LWP changes are higher in BRC_CNST, both PRECL and PRECC changes seem to differ less between the two model runs. This emphasizes the nonlinearity between cloud cover and the microphysical conditions necessary to generate precipitation.

Lastly, the global surface temperature change between the two models is similar and small (BRC: -0.008±0.01 ˚K; BRC_CNST: -0.003±0.01 ˚K).

### 3.6 Effects of BrC bleaching on REari and REaci

Figure 13a shows the REari and REaci due to BrC with a bleaching parameterization from the BRC_BL model run. BrC bleaching significantly reduces REari from 0.13 ± 0.01 W m$^{-2}$ (Fig. 7a) to 0.06 ± 0.008 W m$^{-2}$ (Fig. 13a). What is readily apparent when comparing Fig. 13a to Fig. 7a is that the extent of BrC absorption is greatly constrained when BrC bleaching is incorporated in the model, especially in the tropics (Table 3). This is related to greater insolation in the tropics, leading to more OH formation and a more rapid BrC bleaching process (Eq. (4)). Also, the direct effects associated with BF BrC are greatly reduced with bleaching such that significant effects over most of Asia do not occur.

BRC_BL REaci (Fig. 13b) shows some similar characteristics to BRC REaci (Fig. 7b). Some similar regions with negative REaci include the northeastern Pacific, the south Australian coast, western South America, and near the southern tip of South America. Similar positive REaci regions include Australia, the Weddell Sea, parts of South America, the Gulf of Mexico, and off the coast of southern Africa. In most cases REaci due to aged BrC are smaller in magnitude than the REaci without BrC aging. A notable exception to this is a strong warming signal in Fig. 13b over eastern China (~3 W m$^{-2}$). This correlates with a reduction in LWP over this region (-5 g m$^{-2}$; Fig. S10g) which may also be related to enhanced atmospheric downwelling in this region (Fig. S11b). This indicates some of the nonlinear effects of BrC absorption reduction in the model simulations.

### 3.7 Effects of changing OA density on REari and REaci

Another experiment to determine model sensitivity is a comparison of BRC_CNST model runs with altered input OA density for BB and BF (Table 5). Species density is accessed in the model from the offline phys_prop files and is used to calculate the modal refractive index (Ghan and Zavari, 2007) as well as calculate the number concentration of aerosols activated as cloud condensation nuclei (CCN) (Abdul-Razzak and Ghan, 2000). This is one of the few aerosol species physical properties used directly in the model code, as most of the aerosol physical properties are calculated for each aerosol mode. The inspiration for this comparison lies in the different values used by other modeling studies for OA density, ranging from 1 g cm$^{-3}$ to ~1.5 g cm$^{-3}$. This comparison only looks at the densities used in the default CAM phys_prop file (from Hess et al., 1998) and the density of BB/BF OA used in Feng et al. (2013). Each model run is for the year 2003 and the REari and REaci are both 2003 annual averages.

By decreasing OA density, there is an increase in BrC REari. This is likely due to the use of inverse aerosol density (specific volume) when calculating the radiative effects of the internally mixed aerosol, with a smaller density indicating a larger specific volume and larger BrC absorption attributed to the mixed aerosol. Furthermore, given the same mass concentration and size distribution, an increase in density would reduce the number of particles per volume of air. This would reduce the aerosol optical depth and the RE. In Table 5, the sign on REaci changes dramatically depending on the density used, and is attributed to the alteration of the size of the modal aerosol and its droplet activation efficiency, as well as the number of particles at a given size distribution. The driving factor behind this difference is likely the number of particles per volume of air, which would result in a reduced cloud optical depth with fewer particles (i.e., higher density). A higher density aerosol will also lead to a smaller modal aerosol size, requiring a higher critical supersaturation, which results in lower activation of CCN and less cooling attributed to clouds. What this brief analysis indicates is that there is a need to better constrain OA physical properties if there are to be direct comparisons between simulations of OA radiative effects.

## 4 Discussion and Conclusions

This study has shown that a BrC parameterization in CAM5.4 can bring about significant global radiative effects in CESM. Recent work acting to detail the wavelength dependent optical properties of BrC and its sources has led to its inclusion in chemical transport models and now Earth system models. This is an improvement to the models, which typically assume that OA is not a strong absorber of solar radiation. Furthermore, the incorporation of BrC in CESM allows for analysis of the semi-direct effects of BrC.

One of the first notable findings of this study is that incorporating BrC in CESM exacerbates underestimation of SSA in BB regions compared to AERONET observations. Model underestimation of SSA was found in Jiang et al. (2016) with the default CAM5.3 model. While also noting that the model underestimates BB AOD in the majority of the sites, Jiang et al. (2016) hypothesized that the underestimation of SSA may be due to the model emitting a lower ratio of scattering aerosols to absorbing aerosols than observed. In addition to the possibility of incorrect emission ratios, the model may also be emitting BB particles that are too large (Yu et al., 2016), which could increase absorption efficiency and decrease SSA.

There is also evidence that emission inventories such as GFED 3.1 may underestimate small fire emissions (Werf et al., 2010; Kaiser et al., 2012), leading to model underestimation of absorption in these regions and compounding the model disagreement. Lastly, error introduced from the AERONET data processing for AOD measurement and SSA calculation may explain some of the disagreement between model and observations. Identifying the cause of this problem would be an important step in improving the model in the future given that this stronger absorption is noted in both default and modified CAM code.

Where the BrC incorporation does improve the model compared to AERONET is in its AAE. This emphasizes that the incorporation of BrC in the model better represents aerosol optical properties observed in biomass burning regions. This comparison also revealed strong disagreement between CAM5.4 and AERONET AAE over the Ascension Island site. This may be a factor of size distributions simulated in the model aging processes or of the AERONET cloud screening algorithms filtering out thick smoke days and requires further study. When comparing Mason et al. (2018) observations from Ascension Island, BRC_BL AAE (1.25) has the best agreement to observations (1.16). Other observations show an underestimation of AAE in the model over the US, with improvement noted in the BrC model runs.

The model radiative effect calculations describe an overall warming effect due to BrC in the model as well as regional cooling and warming associated with the BrC semi-direct effect. The global BrC DRE (i.e., REari) is $0.13 \pm 0.01$ W $m^{-2}$. This value is very similar to the Feng et al. (2013) strongly absorbing BrC DRF of $+0.11$ W $m^{-2}$ and the Saleh et al. (2015) core-shell mixing state BrC DRE of $+0.12$ W $m^{-2}$. This effect is not negligible and is equivalent to more than half the BC DRE in CESM (0.25 W $m^{-2}$; Jiang et al., 2016). That being said, incorporation of BrC bleaching in the model led to a significant reduction in globally averaged REari ($0.06 \pm 0.008$ W $m^{-2}$). This value is quite similar to BrC DRE calculated with the same bleaching parameterization in the chemical transport model GEOS-Chem by Wang et al. (2018) (0.05 W $m^{-2}$). The range in our in BrC REari is from 0 to 0.13 W $m^{-2}$, representing the effect of emission of immediately bleached BrC (NOBRC) to no bleaching (BRC). Recent laboratory and observational studies have consistently shown BrC bleaching (Zhong and Jang, 2011; Zhong and Jang, 2014; Lee et al., 2014; Forrister et al., 2015; Zhao et al., 2015; Wang et al., 2016), albeit with varying rates. More work needs to be done to validate bleaching parameterizations in global climate models, but the consistency in observations and the improvement of model performance with BrC bleaching (Wang et al., 2018) supports the inclusion of this parameterization in CESM. Furthermore, this study and that done by Wang et al. (2018) indicate that this process can lead to large differences in BrC direct radiative effects, indicating a smaller global effect due to BrC than has been previously reported in modelling studies.

Looking at the BrC semi-direct effects, the presence of BrC led to regional differences in REaci that spanned from about -2 W $m^{-2}$ to 2.5 W $m^{-2}$, with an overall REaci of $0.01 \pm 0.04$ W $m^{-2}$ for the BRC model run. The REaci due to BrC with a bleaching parameter (i.e., BRC_BL model run) had a similar distribution, but many regions exhibited REaci of smaller magnitude and some regions exhibited REaci of larger magnitude. The global REaci for bleached BrC was very similar to the BRC model with a global average of $0.009 \pm 0.04$ W $m^{-2}$. These effects in both models may be due in part to increased lower tropospheric stability and increased relative humidity (–REaci), decreased entrainment aloft (–REaci), increased static

stability and reduced convection (+REaci), and evaporation of cloud liquid due to absorbing cloud nuclei (+REaci) (Johnson et al., 2004; Sakaeda et al., 2011). However, perturbations in $\omega$ (Fig. 10j, 11j, 12b, S11b) suggest that the annual average REaci are due in large part to altered global circulations due to BrC. Changes in cloud cover, liquid water path, precipitation, and surface flux are also apparent with inclusion of BrC and are correlated with regional changes in REaci in some cases.

However, the mechanisms behind BrC REaci in the model are still unclear due to the complex feedbacks leading to these changes, and nonlinear effects are present with a change in the distribution and magnitude of BrC absorption with the BrC bleaching parameter. Some of these uncertainties may be minimized with a nudged model run that minimizes the sensitivity of the model dynamics to the presence of BrC.

The impact of BrC deposition on snow albedo is not taken into account in this study and should be included before

BrC REsac is considered. Another improvement to the model would be to incorporate this new BrC parameterization into the Snow, Ice, and Aerosol Radiative (SNICAR) model. The SNICAR model has been used in the CESM CLM to diagnose the BC-on-snow effect. This radiative effect has to do with the reduction in albedo of the snow surface due to deposited BC (Flanner et al., 2007). However, OA on snow is not treated in the SNICAR model, and studies have shown that BrC can have an important regional contribution to snow albedo reduction (Doherty et al., 2010).

In comparing the different methods of BrC implementation, BRC and BRC_CNST, we see similar behavior in the model. In Fig. 4, SSA agrees well between the two methods, with the exception of Arctic regions where BRC_CNST predicts slightly stronger absorption. This is likely due to a higher BC-to-OA ratio assumed for BRC_CNST than is seen in the fire emissions in the Arctic. One issue regarding the BRC_CNST implementation is that the model may be overestimating the effect of BrC in the sensitive Arctic regions due to its assumption of a higher BC-to-OA ratio than is

typically observed.

This study also addresses different OA densities used in aerosol modeling studies, and how density affects the RE of BrC. The differing values used in the literature describe the uncertainty in prescribing density to OA. We argue that the variations in this value can lead to significant changes in aerosol direct radiative effect as well as aerosol cloud interactions. A density of 1 g cm$^{-3}$ can result in a 0.02 W m$^{-2}$ higher global average REari than a density of 1.568 g cm$^{-3}$. Changes in cloud

microphysics as a result of changing aerosol density also have an effect on the radiative balance in the model, changing the sign of REaci from negative (1 g cm$^{-3}$) to positive (1.568 g cm$^{-3}$). Due to these effects it seems that deciding on OA physical properties would improve inter-model comparison of BrC radiative calculations.

The BrC improvements introduced in this study are a step towards increasing the accuracy of climate models and the information that they provide. There is also room to improve the accuracy of data sets that feed the model. For example,

work done on improvement of pre-industrial data sets is suggesting that BB emissions were higher and spatially different than pre-industrial data sets currently report (Douglas Hamilton, personal communication). This change would alter calculations of RF that have been done up to now, describing a different anthropogenic effect on the climate than our current understanding. Also, observational datasets looking at vertical distribution of BrC in the atmosphere would help to determine

whether the model is simulating similar processes to observations. This includes more information regarding the transport of BrC to upper levels by deep convection, and the in-cloud aqueous production of BrC (Zhang et al., 2017). GFED emission inventory accuracy is also important because the reported fuel-type and location play a role in the model vertical injection heights of carbonaceous aerosols. More observations of BrC bleaching would help refine the bleaching parameterization used in this study by determining if there are geographic differences in bleaching effect due to differences in solar irradiance. Lower BrC bleaching rates in the Arctic suggest important contributions from BrC deposition on snow. Including more measurements of the radiative effects of BrC impurities in snow could help in the validation of future models that include this surface effect. Lastly, measurements of combustion and non-combustion sources of BrC SOA, as well as their composition/evolution, could aid in the development of BrC SOA in CAM. These and the previously mentioned improvements to this study are steps towards improving CESM and better understanding and predicting the effect that BrC has in the Earth system.

**5 Data availability**

The fire emission data were obtained from the Global Fire Emissions Database (GFED, http://www.globalfiredata.org), and AERONET data were obtained from http://aeronet.gsfc.nasa.gov. Output files from the model runs are available on request from the corresponding authors. Source codes and model setups needed to repeat all simulations are also available upon request.

*Acknowledgements.* This work was supported by the Environmental Protection Agency (EPA) Grant No. 83588301 as well as the Office of Science of the US Department of Energy (DOE) as the NSF-DOE-USDA Joint Earth System Modeling (EaSM) Program and the Earth System Modeling Program. The authors would like to acknowledge the use of computational resources (ark:/85065/d7wd3xhc) at the NCAR-Wyoming Supercomputing Center provided by the National Science Foundation and the State of Wyoming, and supported by NCAR's Computational and Information Systems Laboratory. The authors would like to thank Manvendra Dubey and Dian Romonosky from Los Alamos National Laboratory (LANL) for sharing their biomass burning optical property data for model validation purposes.

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

**Table 1**: Description of the model simulations used in this study

| Model Run | Simulation type | Ensembles | Description |
|---|---|---|---|
| NOBRC | Free-running | 5 | Default CAM5.4 |
| BRC | Free-running | 5 | Modified CAM5.4 with the BB/BF BrC parameterization with changing BC-to-OA ratio [Saleh et al., 2014] |
| BRC_CNST | Free-running | 5 | Modified CAM5.4 with BB/BF BrC parameterization with constant BC-to-OA ratio of 0.08 |
| BRC_BL | Free-running | 5 | Same as BRC but with an additional bleaching parameterization |

| Observation Site | Observation | NOBRC | BRC | BRC_CNST | BRC_BL |
|---|---|---|---|---|---|
| Ascension Island June-October, 2016 (Zuidema et al., 2018) | 1.08 | 1.01 | 1.44 | 1.46 | 1.26 |
| Ascension Island June, 2016 (Zuidema et al., 2018) | 1.16 | 1.01 | 1.43 | 1.45 | 1.25 |
| Las Conchas Fire, NM July, 2011 (Liu S. et al., 2014) | 2.08 (0.6-5.74) | 1.17 | 1.37 | 1.39 | 1.3 |
| Prescribed burns and wildfires, US August-September, 2013 (Mason et al., 2018) | 2.2 (1.9-2.5) | 1.03 | 1.19 | 1.19 | 1.11 |

**Table 2:** Comparisons between modelled AAE and AAE from observations at Ascension Island during the LASIC (Layered Atlantic Smoke Interactions With Clouds (LASIC) campaign, the Las Conchas fire in New Mexico, and biomass burning and prescribed burns from the US during SEAC[4]RS (Studies of Emissions and Atmospheric Composition, Clouds and Climate Coupling by Regional Surveys).

**Table 3**: Different RE contributions globally, in the tropics (25˚S to 25˚N), and in the Arctic (60˚N to 90˚N) from BRC, BRC_CNST, and BRC_BL model runs. These values are from a 9-year period, 2003-2011. Standard deviations from the 5-ensemble means are included.

| BrC RE | BRC | BRC_CNST | BRC_BL |
|---|---|---|---|
| **Global** | | | |
| Aerosol-radiation interaction (REari; W m$^{-2}$) | $0.13 \pm 0.01$ | $0.13 \pm 0.01$ | $0.06 \pm 0.008$ |
| Aerosol-cloud interaction (REaci; W m$^{-2}$) | $0.01 \pm 0.04$ | $-0.001 \pm 0.05$ | $0.009 \pm 0.04$ |
| **Tropics (25˚S to 25˚N)** | | | |
| Aerosol-radiation interaction (REari; W m$^{-2}$) | $0.14 \pm 0.01$ | $0.14 \pm 0.02$ | $0.05 \pm 0.014$ |
| Aerosol-cloud interaction (REaci; W m$^{-2}$) | $0.04 \pm 0.11$ | $0.03 \pm 0.1$ | $0.04 \pm 0.05$ |
| **Arctic (60˚N to 90˚N)** | | | |
| Aerosol-radiation interaction (REari; W m$^{-2}$) | $0.21 \pm 0.04$ | $0.23 \pm 0.02$ | $0.14 \pm 0.04$ |
| Aerosol-cloud interaction (REaci; W m$^{-2}$) | $0.18 \pm 0.3$ | $-0.02 \pm 0.15$ | $0.19 \pm 0.18$ |

**Table 4**: Global averages over the period 2003-2011 for select climate variables from BRC, BRC_CNST, and BRC_BL. Standard deviations from the 5-ensemble means are included.

| Climate Variable | BRC | BRC_CNST | BRC_BL |
| --- | --- | --- | --- |
| Atmospheric absorption (W m$^{-2}$) | $0.29 \pm 0.01$ | $0.28 \pm 0.03$ | $0.13 \pm 0.02$ |
| Surface solar flux (W m$^{-2}$) | $-0.15 \pm 0.02$ | $-0.18 \pm 0.06$ | $-0.08 \pm 0.06$ |
| Low-level cloud fraction (LCF; %) | $0.03 \pm 0.02$ | $0.06 \pm 0.03$ | $0.04 \pm 0.03$ |
| Mid-level cloud fraction (MCF; %) | $-0.02 \pm 0.02$ | $-0.006 \pm 0.02$ | $-0.03 \pm 0.02$ |
| Convective precipitation (PRECC; mm day$^{-1}$) | $-0.007 \pm 0.003$ | $-0.004 \pm 0.001$ | $-0.003 \pm 0.003$ |
| Large-scale precipitation (PRECL; mm day$^{-1}$) | $0.0003 \pm 0.001$ | $-0.0007 \pm 0.003$ | $0.002 \pm 0.003$ |
| Liquid water path (LWP; g m$^{-2}$) | $0.004 \pm 0.1$ | $0.06 \pm 0.09$ | $0.003 \pm 0.13$ |
| Surface temperature (TS; ˚K) | $-0.008 \pm 0.01$ | $-0.003 \pm 0.01$ | $0.004 \pm 0.014$ |

**Table 5:** Comparison of different model runs for the year 2003 with varying OA density. The values are BB and BF OA density from Feng et al (2013), and the default CAM OA properties (Hess et al. (1998). REari and REaci are annual averages calculated for the BRC_CNST model run.

|  | **Density** | REari | REaci |
|---|---|---|---|
| Feng et al. (2013) > | **1.568 g cm$^{-3}$** | 0.14 W m$^{-2}$ | 0.09 W m$^{-2}$ |
| Default CAM > | **1.000 g cm$^{-3}$** | 0.16 W m$^{-2}$ | -0.20 W m$^{-2}$ |

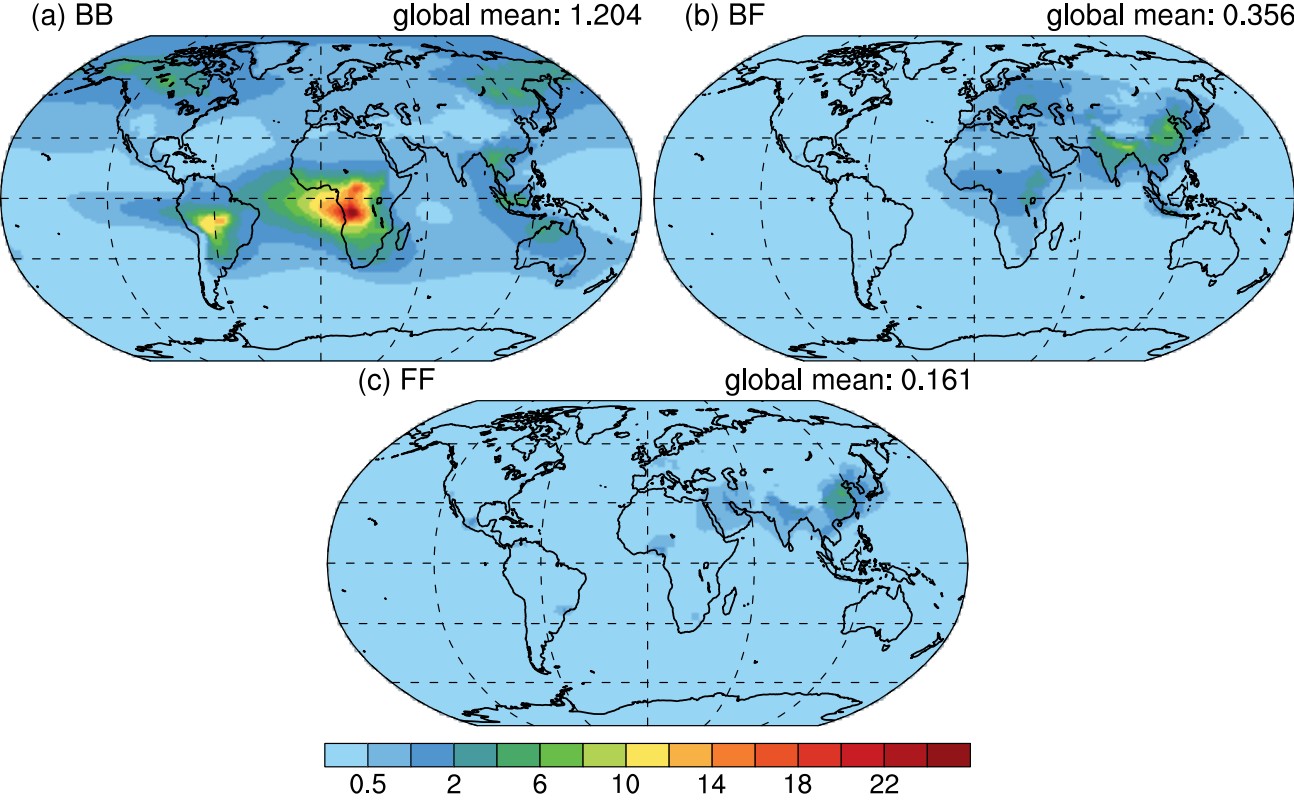

**Figure 1**: The column burdens of POM from (a) biomass burning (BB), (b) biofuel (BF), and (c) fossil fuel emissions. Panels (a) and (b) represent BrC in the BRC, BRC_CNST, and BRC_BL model runs. The units are in mg m$^{-2}$ and the values are a 9 year average from 2003-2011.

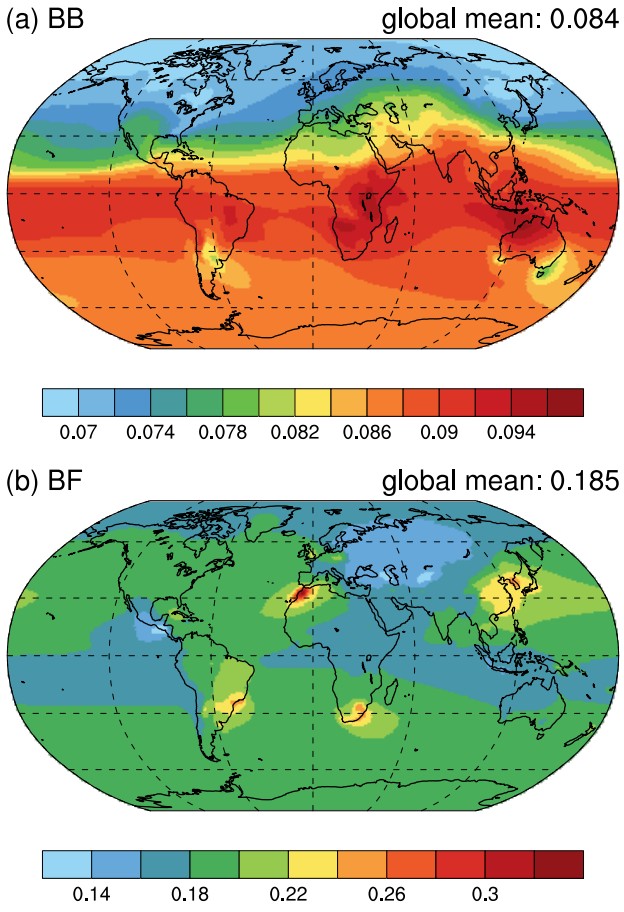

**Figure 2**: The ratio of column averaged burden of black carbon to POM of (a) biomass burning (BB) and (b) biofuel burning (BF). Note the different scales.

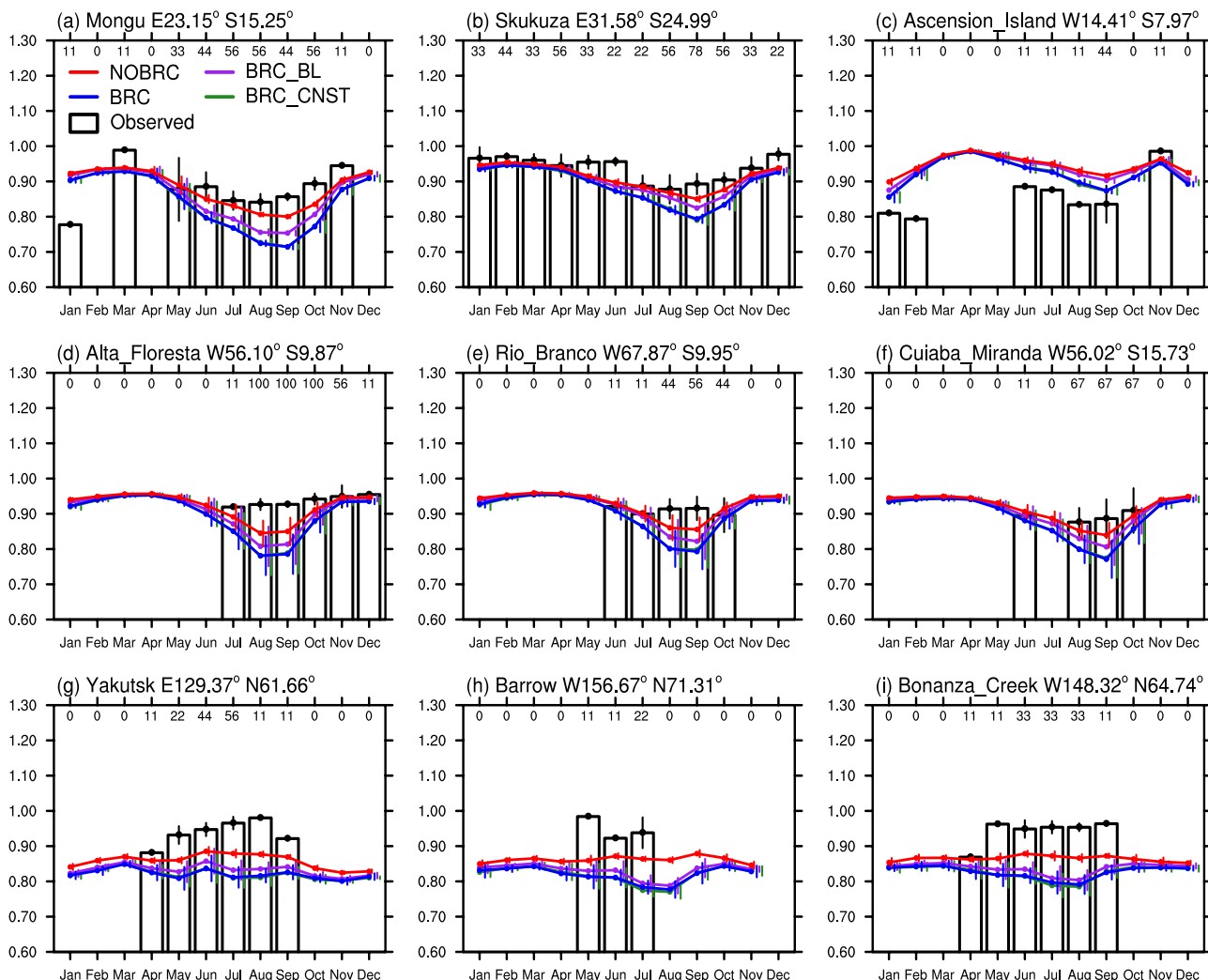

**Figure 3**: AERONET SSA compared to CAM5.4 model output SSA, not including brown carbon (NOBRC), including brown carbon (BRC, BRC_CNST), and including brown carbon and a bleaching effect (BRC_BL). Vertical lines are color-coded error bars and run from left to right: Observed, NOBRC, BRC, BRC_BL, BRC_CNST. Values under the upper x-axis indicate percentage of available data in the 9-year period.

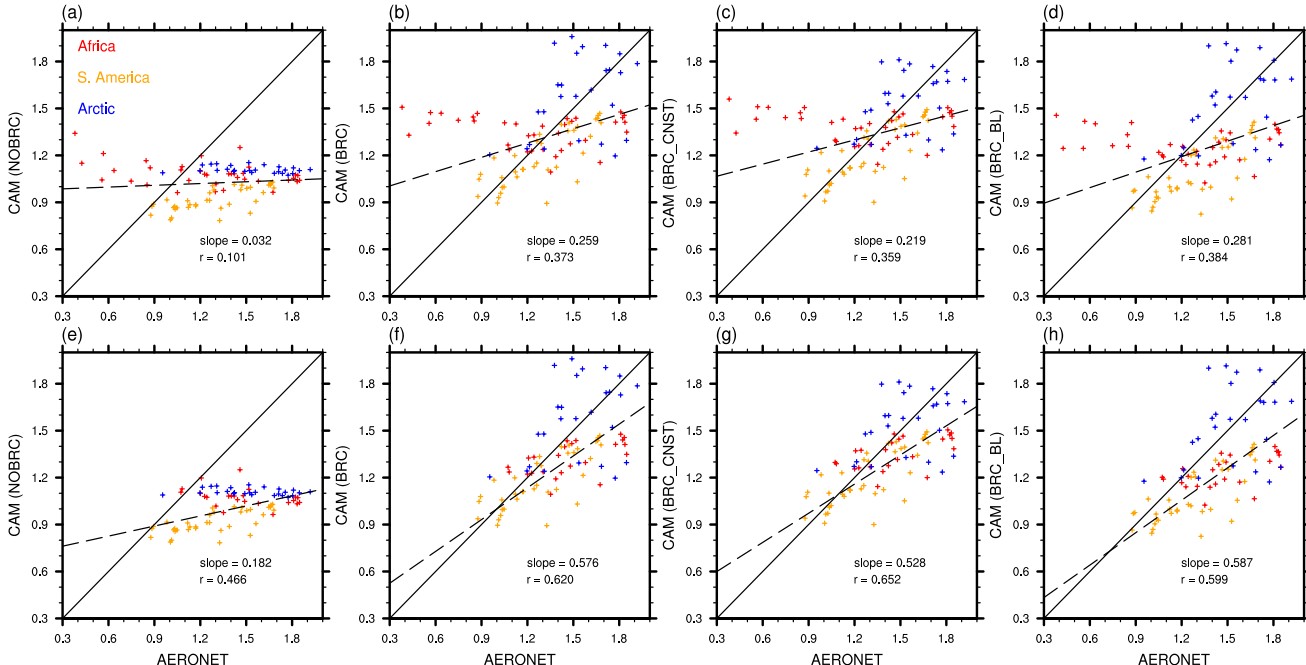

**Figure 4:** Absorption Angstrom exponent (AAE) comparison between all of the AERONET sites from Fig. 4 and the four model runs: (a) NOBRC, (b) BRC, (c) BRC_CNST, and (d) BRC_BL. The lower row looks at the same model comparisons neglecting the Ascension Island site. Red indicates Africa sites, orange indicates South American sites, and blue indicates Arctic sites. The solid line indicates a 1:1 fit and the dotted line is a best fit to the data with the slope and correlation coefficient shown.

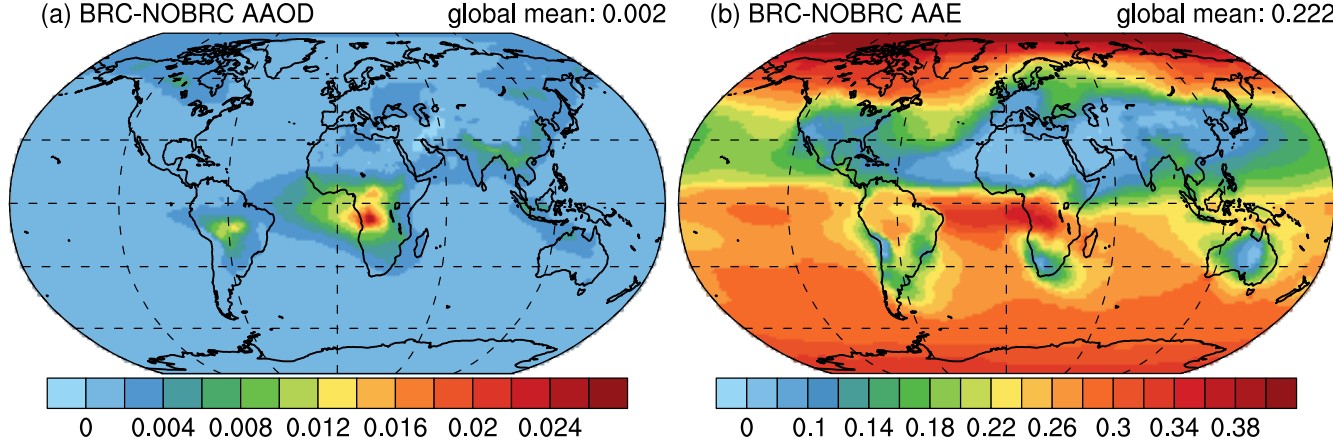

**Figure 5**: Differences in (a) AAOD and (b) AAE between the BRC model run and the NOBRC model run.

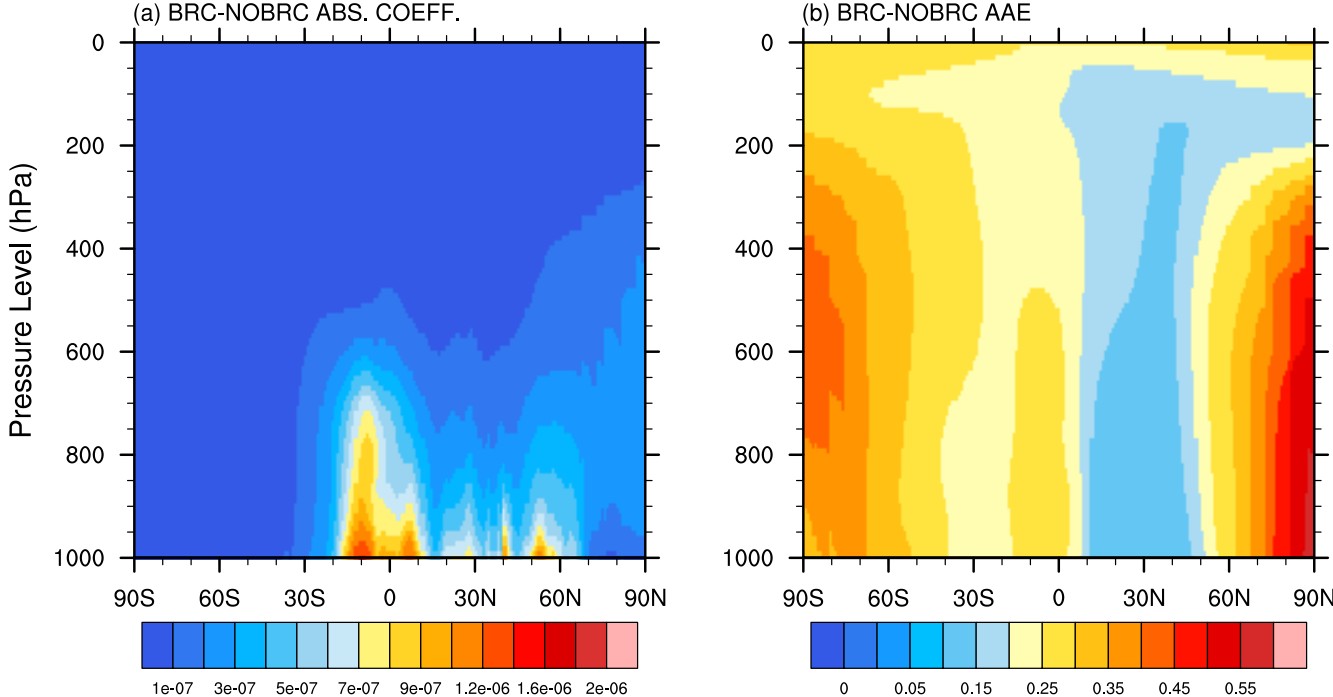

**Figure 6**: Differences in (a) zonally averaged absorption coefficient (m$^{-1}$) and (b) zonally averaged AAE between the BRC model run and the NOBRC model run.

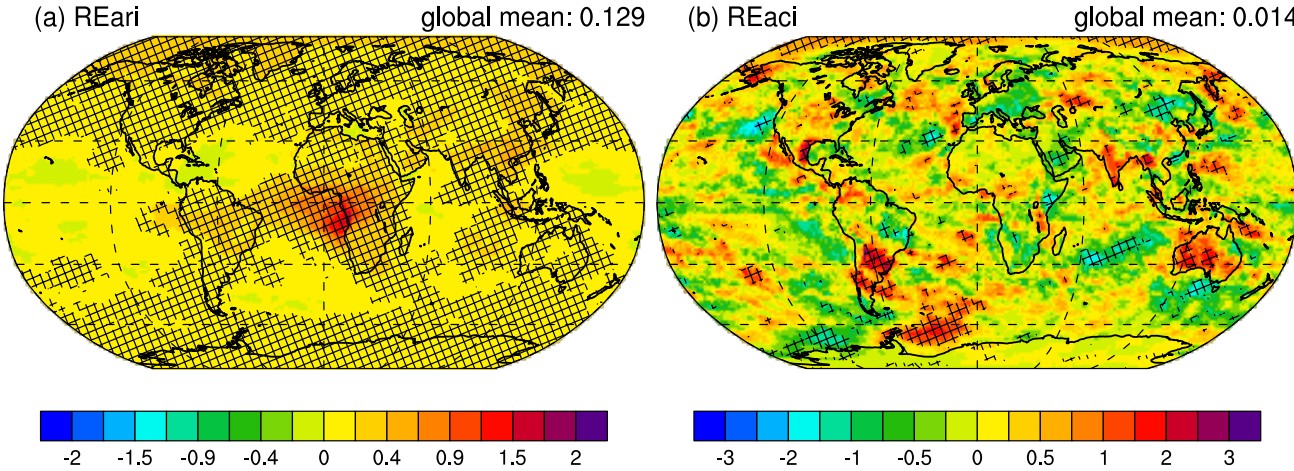

**Figure 7**: Radiative effects from (a) aerosol-radiation interactions (REari) and (b) aerosol-cloud interactions (REaci) from the BRC model run. Units are W m⁻². Note different scales. Hatching indicates regions where the change over the ensemble years is significant to the 0.05 (REari) and 0.1 (REaci) levels.

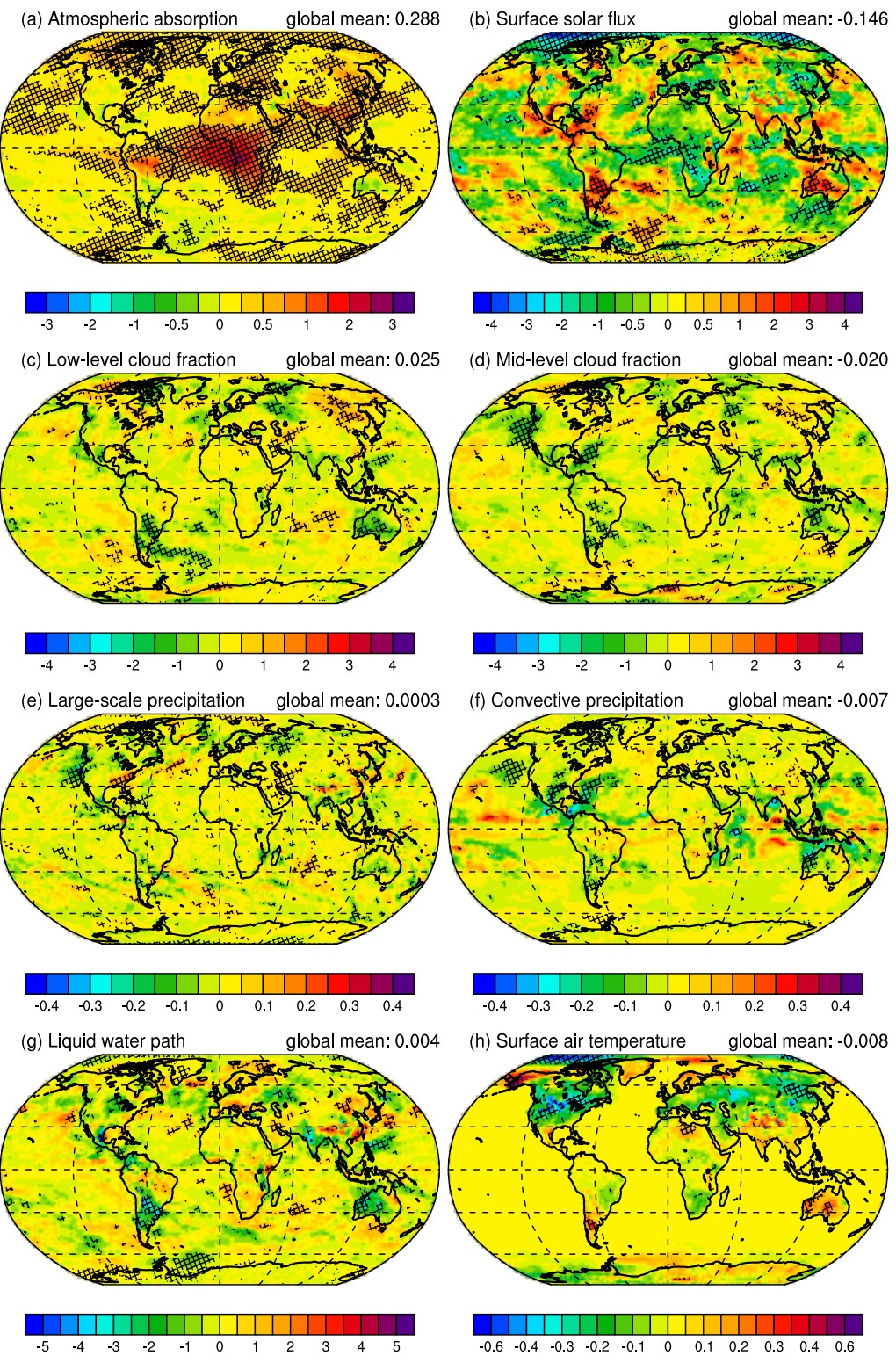

**Figure 8**: The effect of BrC addition in the BRC model run on (a) atmospheric absorption (W m$^{-2}$), (b) surface solar flux (W m$^{-2}$), (c) low-level cloud fraction (%; clouds below 700 mb), (d) Mid-level cloud fraction (%; clouds between 700 and 400 mb), (e) large-scale precipitation (mm day$^{-1}$), (f) convective precipitation (mm day$^{-1}$), (g) liquid water path (g m$^{-2}$), (h) surface air temperature (˚K). Hatching indicates significance in ensemble year change to the 0.1 level.

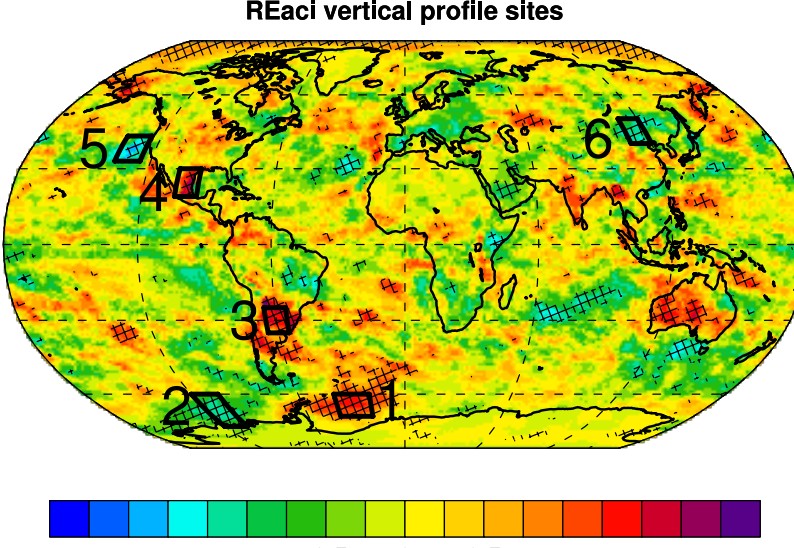

**Figure 9:** Locations for the vertical profile sites influenced by significant REaci in the BRC model run. Sites are referenced herein as: 1) Weddell Sea (WS), 2) Antarctic Coast (AC), 3) South America (SA), 4) Gulf of Mexico (GM), 5) northeastern Pacific (NEP), northeastern China (NEC). Hatching indicates significance in the ensemble year change to the 0.1 level.

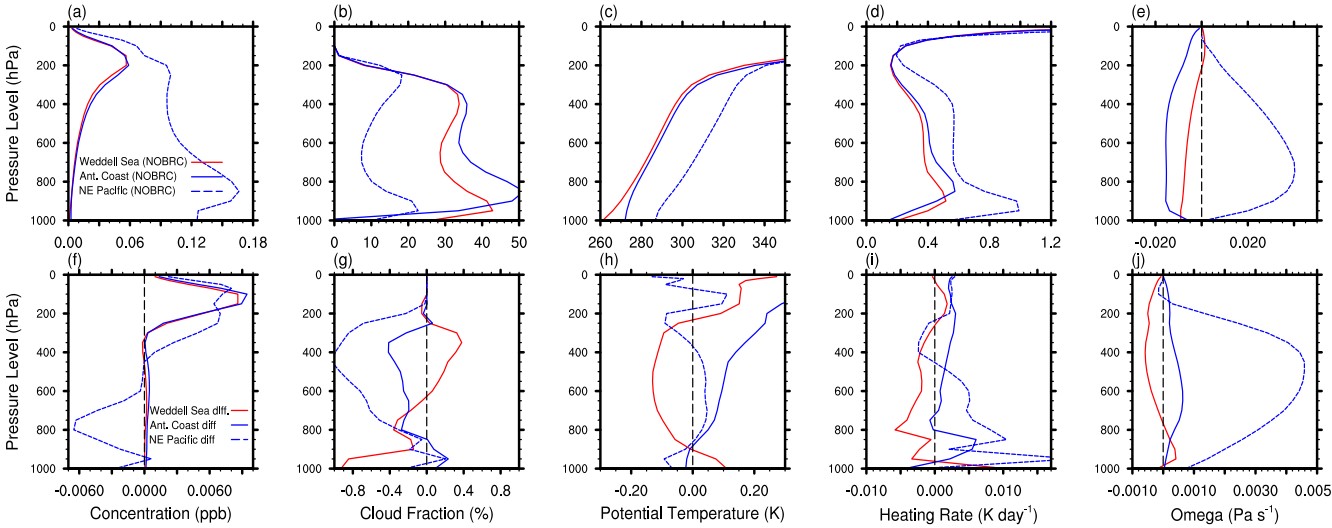

**Figure 10:** Vertical profiles (a-e) and vertical profile perturbations (f-j) for the sites Weddell Sea (WC) and Antarctic Coast (AC). In plots a-e the model run NOBRC is plotted, and the difference between BRC and NOBRC is plotted in f-j. Red represents regions with a positive REari and blue represents regions with a negative REari.

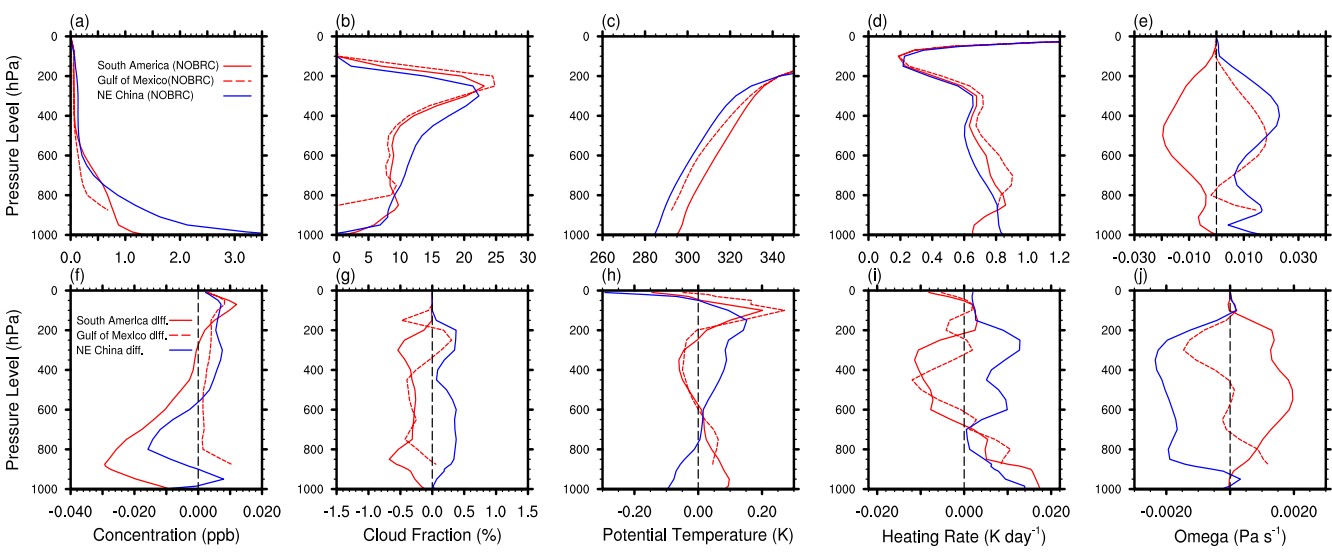

5    **Figure 11:** Vertical profiles (a-e) and vertical profile perturbations (f-j) for the sites South America (SA), Gulf of Mexico (GM), and Northeastern China (NEC). In plots a-e the model run NOBRC is plotted, and the difference between BRC and NOBRC is plotted in f-j. Red represents regions with a positive REari and blue represents regions with a negative REari.

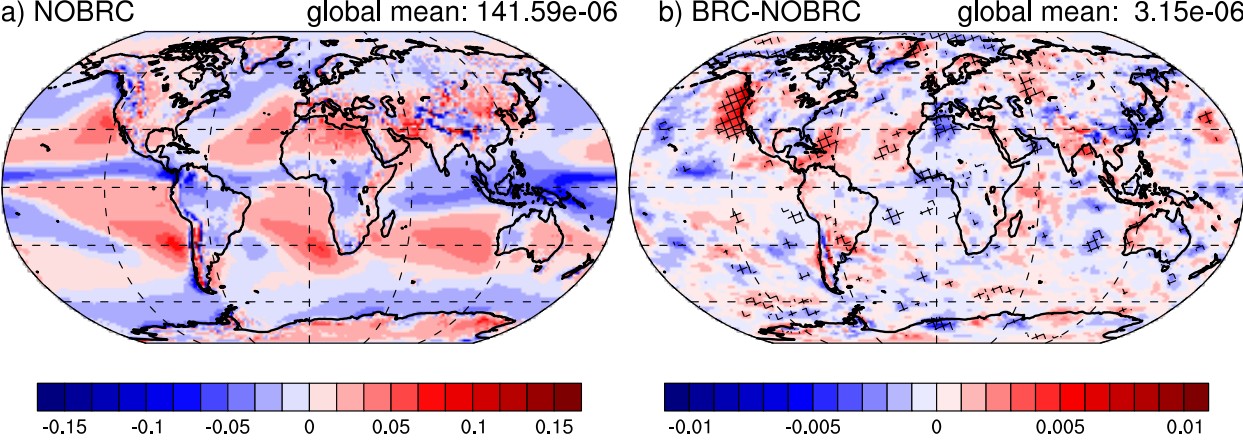

**Figure 12:** 500 mb omega in Pa s$^{-1}$ from the BRC model run: (a) the default annual average and (b) the difference with the incorporation of BrC. Hatching indicates significance in the ensemble year change to the 0.1 level.

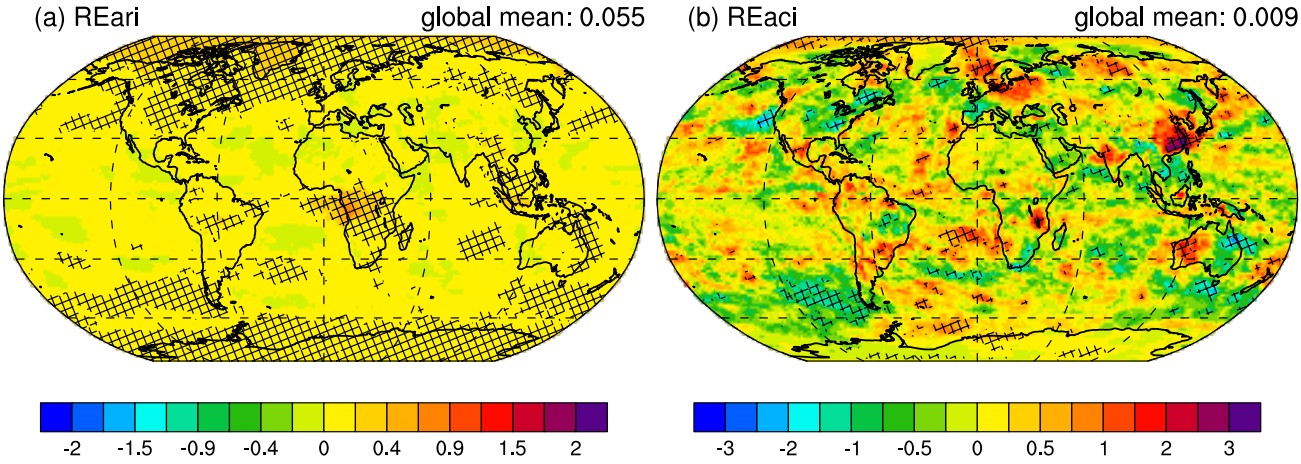

**Figure 13**: Same as Fig. 6 but for the BRC_BL model run. Hatching indicates regions where the change over the ensemble years is significant to the 0.05 (REari) and 0.1 (REaci) levels.