# Peer review of "Radiative Effect and Climate Impacts of Brown Carbon with the Community Atmosphere Model (CAM5)"

_Atmospheric Chemistry and Physics, 2018_

## Referee Comment (RC1) · Anonymous Referee #1 · 28 Aug 2018

This paper advances the understanding of the global radiative impacts of light absorbing PM organic species (BrC). It appears to be the first research to included BrC in an earth system model (CESM), which goes beyond previous models that only considered BrC direct radiative forcing effects. This more advanced model considers factors such as surface albedo, clouds and various atmospheric dynamic processes. By including the important process of BrC bleaching, Wang et al 2018 made a substantial improvement over previous models of BrC global impacts that assumed it was largely invariant once emitted. This model also considers bleaching, (although only resulting from particle reaction with OH), and with added secondary BrC effects, is likely the most advanced to date.

[Figure]

All these models, including the one described in this paper, are still overly simplistic and so the results highly uncertain. The fundamental problem is that not all the processes that influence BrC are known, and there are really no global scale data sets of BrC which can be used to test the model predictions. As has been done in some prior studies, AERONET data are used in this work, but provide only limited validation (inclusion of BrC shows better agreement with AAEs). Because of the advances in modeling BrC over what has previously been done, this paper is a worthwhile contribution, but the specific results are highly uncertain and speculative.

In addition to (or maybe instead of) using a model to simply assess BrC climate impacts that really can't be verified at this point, additional discussion could be added on what the authors feel could be done to help assess various model performance and move research of BrC radiative impacts forward. For example, are there places where measurements of BrC would be most beneficial? The authors could show more detailed spatial distributions (including vertical profiles) of BrC and BC (maybe include mineral dust too). TOA forcing is highly sensitive to the vertical distribution of light absorbers, and there is evidence that BrC can be enhanced at higher altitudes relative to BC, how confident are the authors of the vertical distribution of BrC in their model, how does the model consider vertical transport of BrC, what is the effect of this uncertainty on radiative forcing. Only spatial distributions of POM are shown, similar results for BrC would be of interest. If BrC vertical structure is also important for stability, cloud formation etc, (affects other than direct radiative forcing), what are the limitations with the model in this respect. Another question that may be of interest is how does the model-predicted lifetime of BrC vary geographically? This was alluded to in the paper, but maybe could be expanded more. In summary, maybe the authors could list what are the major uncertainties in their analysis of BrC radiative impacts and what is needed to address them.

Minor comments.

P2, L12: Feng et al did not consider BrC bleaching, so this is likely a large over estimation, which should be noted.

Don't really understand the layout of the first 3 equations. Eq 1 should be something like RI = . . .

P9 L18, typo BRC_CL ??

Fig 6 and associated discussion and in the sections that follow; by specific about the brown carbon included in the model, ie, was it BRC, CRC_CNST or BRC_BL?

The model considers BrC bleaching just due to OH. Is this the only route for bleaching? Please justify. What are the limitations with this assumption?

---

## Referee Comment (RC2) · Anonymous Referee #2 · 30 Aug 2018

Review of "Radiative Effect and Climate Impacts of Brown Carbon with the Community Atmosphere Model (CAM5)" by Brown et al.

This manuscript presents an investigation into the direct and indirect radiative effect form brown carbon using the CESM model. In all, this work is well designed, executed, and of broad interest to the aerosol science community. The writing is clear and thorough. It is worthy to be published on ACP after minor revisions.

Current knowledge of both brown carbon and how aerosols interact with cloud are with large uncertain, therefore I can imagine the uncertainties in this study may be the subject of discussion. While conducting a very accurate analysis is nearly impossible

at this point, I suggest the authors to include a short discussion for the uncertainties. For example, a more reliable range of the radiative effect may be more useful than the global mean numbers. There are also a few questions need to be discussed. How large is the uncertainty? What processes contribute the uncertainty, and what are the most important? What kind of laboratory/field measurements are most useful for reducing the uncertainty?

Specific comments:

- Page 4, line 12: I guess it is 25% instead of 0.25%.

- Page 5, line 10: Could the authors give more information about the aerosol size distributions? What are the median sizes and standard deviations for each mode? Is there microphysical process changing the size in the model? Please also provide the information of BrC density here even it is discussed later.

- Page 5, line 18: does it mean the model assumes totally internal mixing everywhere? The influence of mixing assumption worth a discussion in later sections.

- Page 5, line 27: Is there any reason to use GFED3.1 instead of the current version of GFED4?

- Page 6 the first equation is not clear.

- Page 6, line 24: I cannot understand how the BC/OA ratio is used in this parameterization. Is the BC/OA ratio calculated from the emissions for every grid box at every timestep? Does this mean you assume all the BrC currently simulated in the grid box has the same absorption property as those emitted locally?

- Page 8, line 20: What version of AERONET data is used? level 1.5 or level 2? How good are the AERONET data? How many data points do you have (% per season)?

- Page 8, line 24: I cannot understand why the authors used 550nm data in their analysis. 440nm is a much better choice to evaluate BrC.

[Figure]

- Figure 3 is not that useful for the readers, maybe combine Figure 3 and 4 to an absorption AOD plot.

- Page 9, line 28: what wavelengths are $\lambda 1$ and $\lambda 2$?

- Before talking about the BrC effect. It may be worthy to describe the model result of BrC absorption briefly. For example, global mean absorption AOD and AAE, spatial and vertical variations of BrC absorption, contribution to total aerosol absorption, etc.

- There are a lot of measurements of BrC absorption in the literature (most at the surface). A brief comparison between model and these observations could provide useful information beyond the limited model validation by AERONET.

---

## Author Comment (AC2) · 12 Nov 2018

**Responses to all reviewer's comments. The reviewer comments are italicized and our responses are not.**

**Reviewer #2**

*This manuscript presents an investigation into the direct and indirect radiative effect form brown carbon using the CESM model. In all, this work is well designed, executed, and of broad interest to the aerosol science community. The writing is clear and thorough. It is worthy to be published on ACP after minor revisions.*

Reply: We thank the reviewer for the positive and encouraging comments.

**1.** *Current knowledge of both brown carbon and how aerosols interact with cloud are with large uncertain, therefore I can imagine the uncertainties in this study may be the subject of discussion. While conducting a very accurate analysis is nearly impossible at this point, I suggest the authors to include a short discussion for the uncertainties. For example, a more reliable range of the radiative effect may be more useful than the global mean numbers.*
Reply:
We appreciate the reviewer's comment about increased discussion regarding model uncertainty. In regards to a more detailed range in BrC RE, sections 3.2 and 3.6 detail regions of significant REari and REaci. We feel that this gives a good representation of the global range in BrC RE. Furthermore, the three model runs NOBRC, BRC_BL, and BRC give a range of 100% immediate bleaching of BrC to no bleaching of BrC. We mention this in the discussion.

We made a change on **page 19, line 20** – "The range in our in BrC REari is from 0 to 0.13 W m$^{-2}$, representing the effect of emission of 100% immediately bleached BrC (NOBRC) to no bleaching (BRC)."

We added a discussion for the uncertainties in BrC RE (see responses to the reviewer's comment 2 below).

**2.** *There are also a few questions need to be discussed. How large is the uncertainty? What processes contribute the uncertainty, and what are the most important? What kind of laboratory/field measurements are most useful for reducing the uncertainty?*
Reply:
We have taken special care to include more information regarding the uncertainty in the implemented parameterizations as well as in the model set-up (sections 2.2.2 and 2.2.3). We also included a discussion of information needed to minimize uncertainty in the model at the end of the discussion.

[revised manuscript text omitted]

*Specific comments:*

**3.** *Page 4, line 12: I guess it is 25% instead of 0.25%.*
Reply:
We have fixed this value.

**4.** *Page 5, line 10: Could the authors give more information about the aerosol size distributions? What are the median sizes and standard deviations for each mode? Is there microphysical process changing the size in the model? Please also provide the information of BrC density here even it is discussed later.*
Reply:
We included a better description of the median size distributions and included the standard deviations for each mode. We also mentioned in more detail how the aging process converts Primary Carbon mode to Accumulation mode, increasing the size and hygroscopicity of the primary carbon mode as it ages. We added a mention of BrC density in section 2.2.2 when we discuss the BrC parameterization in the model.

We made a change on **page 5, line 9** - This model also uses the 4-mode version of MAM (MAM4) (Liu X. et al., 2016). MAM4 consists of the following four lognormal modes (shown with their median size ranges and standard deviations): Aitken (0.015 – 0.053 $\mu$m, $\sigma = 1.8$), accumulation (0.058 – 0.27 $\mu$m, $\sigma = 1.6$), coarse (0.80 – 3.65 $\mu$m, $\sigma = 1.8$), and primary carbon (0.039 – 0.13 $\mu$m, $\sigma = 1.6$). The median sizes of aerosol modes are changed due to the microphysical processes (e.g., condensation and coagulation) while standard deviations for each mode are fixed. The OA from accumulation and primary

carbon modes, which is used to represents BrC depending on its source (mentioned later), has a density in the model of 1 g cm$^{-3}$.

**5.** *Page 5, line 18: does it mean the model assumes totally internal mixing everywhere? The influence of mixing assumption worth a discussion in later sections.*
Reply:
The model assumes internal mixing within the mode and external mixing between the modes when calculating the optical properties of each aerosol mode. We mention the impact that this treatment may have on the radiative effects in the model when discussing uncertainties

We made a change on **page 5, line 21** - "The optical calculations in the model consider these aged particles to be internal mixtures of aerosols within the mode, and the RI of the accumulation mode, as well as the other 3 modes, is calculated as a volume-weighted mean of the refractive indices of all of the aerosol's components within the mode (Liu X. et al., 2012; Liu X. et al., 2016)."

We made a change on **page 7, line 16** - "Another assumption is the model use of a volume mixing assumption, which may act to overestimate aerosol light absorption (Jacobson, 2000; Adachi et al., 2011)."

**6.** *Page 5, line 27: Is there any reason to use GFED3.1 instead of the current version of GFED4?*
Reply:
GFED3.1 was used in order to allow comparison to the Jiang et al. (2016) study which focused on the radiative effects of biomass burning aerosols (neglecting BrC). It is a valid point that using this older version differs in several aspects from the most recent emissions dataset, and we add some discussion regarding the effect of this change on the results.

We made a change on **page 6, line 1** - "The GFED 3.1 emissions were used in this study to allow for direct comparison between this study and Jiang et al. (2016). The more recent GFED 4 emission dataset shows an 11% global increase in fire emissions from GFED 3.1 (Werf et al., 2017), which may result in a slightly stronger climate impact from biomass burning aerosols than that shown in this study."

**7.** *Page 6 the first equation is not clear.*
Reply:
Equation (1) has been modified to

$$RI = 1.7(\pm0.2) + k_{OA}i = 1.7(\pm0.2) + k_{OA,550}\left(\frac{550}{\lambda}\right)^{w}i$$

**8.** *Page 6, line 24: I cannot understand how the BC/OA ratio is used in this parameter-ization. Is the BC/OA ratio calculated from the emissions for every grid box at every timestep? Does this mean you assume all the BrC currently simulated in the grid box has the same absorption property as those emitted locally?*
Reply:
That is correct. The BC/OA ratios described in the model run don't change substantially within biomass burning emission plumes, leading us to assume a relatively small effect on the imaginary part of the refractive index. When assuming that high BC/OA emissions from Africa (BC/OA = 0.1) can make it up to the Arctic (BC/OA = 0.06), we find that $k_{OA}$ uncertainty is just under 10%. Given the reduction in aerosol absorption due to wet and dry deposition as well as dilution, the effect of this uncertainty on the radiative effect is further reduced. We address this in our uncertainty discussion and assume that this uncertainty is negligible.

We made a change on **page 7, line 17** - "We also assume that the BC-to-OA ratio in transported smoke is similar to BC-to-OA from the source region, allowing for the use of a BC-to-OA ratio at each gridcell at every time step to calculate $k_{OA}$ in each gridcell. The uncertainty in $k_{OA}$ associated with this assumption is small (<10% for BB emissions assuming transport from the Equator to the Arctic (not shown)) and is assumed to be negligible."

**9.** *Page 8, line 20: What version of AERONET data is used? level 1.5 or level 2? How good are the AERONET data? How many data points do you have (% per season)?*
Reply:
We use level 2.0 AERONET data and add mention of this on page 9, line 25. There is more available AOD data than SSA data due to the uncertainty involved in the inversion algorithm used to calculate SSA in low aerosol environments (Dubikov et al., 2000). We added percentages of data for the 9-year period for each of the months in the AOD, AAOD, and SSA plots, under the upper x-axis.

Modified **Fig. 3,** and added **Figs. S3, S4.**

**10.** *Page 8, line 24: I cannot understand why the authors used 550nm data in their analysis. 440nm is a much better choice to evaluate BrC.*
Reply:
Figures were modified to reflect 440 nm AERONET data.

Modified **Figs. 3, S3, S4.**

**11.** *Figure 3 is not that useful for the readers, maybe combine Figure 3 and 4 to an absorption AOD plot.*
Reply:
We have included an AAOD plot in the supplementary. We did not include this in the main text because the change in AAOD magnitude between the different sites is large, and the large uncertainty in model AAOD makes for a less useful model comparison. We left the SSA plot in the paper and moved the AOD plot to the supplementary.

We added **Fig. S4.**

**12.** *Page 9, line 28: what wavelengths are λ1 and λ2?*
Reply:
λ1 = 440 nm and λ2 = 675 nm. This was added to the paper.

We made a change on **page 10, line 31** - "AAE is calculated based on the two wavelegths 440 nm and 675 nm ($\lambda_1$ and $\lambda_2$, respectively) and the measured absorption coefficients at the two different wavelengths ($b_{abs}(\lambda_1)$ and $b_{abs}(\lambda_2)$)."

**13.** *Before talking about the BrC effect. It may be worthy to describe the model result of BrC absorption briefly. For example, global mean absorption AOD and AAE, spatial and vertical variations of BrC absorption, contribution to total aerosol absorption, etc.*
Reply:
We have added two plots that show the difference in AAOD, AAE, zonally averaged absorption coefficient, and zonally averaged AAE due to the BrC implementation.

We have added **Fig. 5** and **Fig. 6.**

We made a change on **page 12, line 11** - "This can be seen in Fig. 5a which shows the difference in AAOD between the BRC and NOBRC model runs. The max AAOD over southern Africa is about 0.024, which makes up approximately 1/3 of the total AAOD in this same region (Fig. S7a). As with Fig. 4, Fig. 5b shows a global increase in AAE due to BrC. These effects are the strongest over the southern African BB region and in the Arctic, with the strongest AAE increases over the Arctic (Fig. 5b) correlated with weaker AAOD in Fig. 5a. Vertical cross-sections of aerosol absorption coefficient and AAE changes due to BrC (Fig. 6), show the vertical extent of BrC. Zonal BrC absorption is dominated by the African and South American biomass burning regions, with visible aerosol transport to the Arctic from boreal fires (Fig. 6a). While absorption over the Antarctic is nearly zero, upper level transport of dilute BrC to the south can be inferred from the AAE changes in Fig. 6b.."

**14.** *There are a lot of measurements of BrC absorption in the literature (most at the surface). A brief comparison between model and these observations could provide useful information beyond the limited model validation by AERONET.*
Reply:
We include a table of comparison between three different AAE surface measurements: Ascension Island observations from the LASIC (Layered Atlantic Smoke Interactions With Clouds (LASIC) campaign, the Las Conchas fire in NM, and biomass burning and prescribed burns from SEAC⁴RS (Studies of Emissions and Atmospheric Composition, Clouds and Climate Coupling by Regional Surveys). We also mention biomass burning AAE from three 2018 fires in NM.

We added **Table 2.**

[revised manuscript text omitted]

---

## Author Response (AR1)

[revised manuscript text omitted]

Hunter Brown 11/10/18 8:07 PM

Hunter Brown 11/10/18 8:07 PM

Hunter Brown 11/10/18 8:08 PM

Hunter Brown 11/10/18 8:08 PM

Hunter Brown 11/10/18 8:08 PM

Hunter Brown 11/10/18 8:08 PM

Hunter Brown 11/10/18 8:08 PM

**3.4 Vertical changes due to BrC semi-direct effects**

To understand how the BRC model BrC plays a role in aerosol-cloud interactions we looked at vertical profiles of land (western Gulf of Mexico (GM), South America (SA), northeastern China (NEC)) and oceanic regions (Weddell Sea (WS), western Antarctic Coast (AC), northeast Pacific (NEP)) with significant positive and negative BrC REaci (Fig. 9). The vertical profiles come from the averaged grids in these regions with greater than a 0.9 confidence level (Fig. 10 and 11).

In the Antarctic oceanic regions, there is little influence from BB and BF POA (i.e., sources of BrC) (Fig. 10a) and there is a positive vertical motion associated with the Ferrel Cell convergence zone (Fig. 10e). The NEP site has a larger aerosol influence as well as negative vertical motion due to the descending branch of the Hadley circulation. WS low-level cloud fraction (Fig. 10b) and potential temperature (Fig. 10c) correspond with the greater annual average ice fraction in the WS (63% versus 15% off the AC). The heating rate in all three regions corresponds to the local cloud fraction maxima, indicating strong contribution from latent heat release (Fig. 10d). In all of the regions, the upper level concentration of BB and BF POA increases by about 10-15% (Fig. 10f) indicating a change in atmospheric circulation with BrC implementation. This is backed up by positive changes in omega (~20% max) (Fig. 10i) and decreases of ~4% (increases of ~2%) in high (low) level clouds (Fig. 10g). The NEP region shows a decrease in aerosol concentration at lower levels (5%), possibly indicating a strengthening of the Hadley cell down welling, which may change advection of aerosol into the region or inhibit advection due to the high-pressure enhancement. In the WS there is a decrease in low-level clouds (3% max) and an increase in upper level clouds (1%), and off the AC the low level clouds are enhanced (1%) (Fig. 10g). Cloud fraction changes at all three sites can be related to changes induced in potential temperature (Fig. 10h) and heating rate (Fig. 10i). Due to low concentrations of aerosol in these regions the main driving force behind the significant REaci may be due to changes in vertical motion (Fig. 10j).

Increased cloud cover near the surface may lead to the negative REaci off the AC, which is consistent with strengthening upward velocities and can also be seen by a slight significant increase in LCF (Fig. 8c). The factors leading to a positive REaci in the WS may be the increased high cloud cover as well as the decreased low cloud cover correlating with descending air near the surface (Fig. 10j). Why the air descends in this region is unclear, but may be related to the larger ice fraction in the region contributing to an inversion that may decouple the lower levels from the upper level circulation changes as observed in a WS field campaign by Andreas et al. (2000).

The land sites that are selected have a much stronger influence from BB and BF POA than the Antarctic Ocean sites (Fig. 11a). NEC is marked by slightly higher midlevel cloud fractions and lower temperatures than the GM or SA (Fig. 11b-c), and is in a region of descending air aloft (Fig. 11e). Both the GM and SA are located in latitudes near the descending branch of the Hadley circulation but are marked by different regional circulations: SA has an annual average upwelling (Fig. 11e, 12a) influenced by lee-side cyclogenesis (Mendes et al., 2007) while GM has annual average downwelling influenced by the Hadley circulation (Fig. 11e, 12a). Figure 11f shows changes in POA concentrations ranging from ~4% decrease to ~1% increase near the surface. Regions with positive REaci have decreased cloud fraction (~7% max) while regions with

Hunter Brown 11/10/18 8:10 PM
Hunter Brown 11/10/18 8:10 PM
Hunter Brown 11/10/18 8:10 PM
Hunter Brown 11/10/18 8:10 PM
Hunter Brown 11/10/18 8:10 PM
Hunter Brown 11/10/18 8:14 PM
Hunter Brown 11/10/18 8:14 PM
Hunter Brown 11/10/18 8:14 PM
Hunter Brown 11/10/18 8:14 PM
Hunter Brown 11/10/18 8:14 PM
Hunter Brown 11/10/18 8:15 PM
Hunter Brown 11/10/18 8:15 PM
Hunter Brown 11/10/18 8:15 PM
Hunter Brown 11/10/18 8:15 PM
Hunter Brown 11/10/18 8:15 PM
Hunter Brown 11/10/18 8:15 PM
Hunter Brown 11/10/18 8:17 PM
Hunter Brown 11/10/18 8:17 PM
Hunter Brown 11/10/18 8:17 PM

[revised manuscript text omitted]

Hunter Brown 11/11/18 10:41 AM

**Author response to reviewer comments. The reviewer comments are italicized and our responses are not.**

**Reviewer #1**

*This paper advances the understanding of the global radiative impacts of light absorbing PM organic species (BrC). It appears to be the first research to included BrC in an earth system model (CESM), which goes beyond previous models that only considered BrC direct radiative forcing effects. This more advanced model considers factors such as surface albedo, clouds and various atmospheric dynamic processes. By including the important process of BrC bleaching, Wang et al 2018 made a substantial improvement over previous models of BrC global impacts that assumed it was largely invariant once emitted. This model also considers bleaching, (although only resulting from particle reaction with OH), and with added secondary BrC effects, is likely the most advanced to date.*
Reply:
We thank the reviewer for the positive and encouraging comments.

*All these models, including the one described in this paper, are still overly simplistic and so the results highly uncertain. The fundamental problem is that not all the processes that influence BrC are known, and there are really no global scale data sets of BrC which can be used to test the model predictions. As has been done in some prior studies, AERONET data are used in this work, but provide only limited validation (inclusion of BrC shows better agreement with AAEs). Because of the advances in modeling BrC over what has previously been done, this paper is a worthwhile contribution, but the specific results are highly uncertain and speculative.*
Reply:
We thank the reviewer for the positive comments. We have added more comparisons with observational data in addition to AERONET, and discuss the uncertainty of our results in the revised manuscript. See our responses to the specific comments below.

**1.** *In addition to (or maybe instead of) using a model to simply assess BrC climate impacts that really can't be verified at this point, additional discussion could be added on what the authors feel could be done to help assess various model performance and move research of BrC radiative impacts forward. For example, are there places where measurements of BrC would be most beneficial?*
Reply:
We appreciate the recommendation. We added some discussion at the end of the paper regarding measurements that would be beneficial to further development of BrC in the model.

[revised manuscript text omitted]

**3.** *Only spatial distributions of POM are shown, similar results for BrC would be of interest.*
Reply:
In the case of this parameterization, the biomass and biofuel POM tracers represent BrC in the model. We have added a clarification in the caption for Figure 1.

We made a change to the **Fig. 1 Caption** - "The column burdens of POM from (a) biomass burning (BB), (b) biofuel (BF), and (c) fossil fuel emissions. Panels (a) and (b) represent BrC in the BRC, BRC_CNST, and BRC_BL model runs. The units are in mg m$^{-2}$ and the values are a 9 year average from 2003-2011."

**4.** *Another question that may be of interest is how does the model-predicted lifetime of BrC vary geographically? This was alluded to in the paper, but maybe could be expanded more.*
Reply:
We have added a brief analysis of the BrC bleaching effect in different regions of the globe to the supplementary material. We reference this in section 3.1.
As mentioned above, the biomass and biofuel POM tracers represent BrC in the model. Modeled-predicted lifetime of POM is shown in Liu et al. (2016).

Added **Table S1.**

We made a change on **page 10, line 23** - "Table S1 shows [OH] in different regions and the half-life of BrC due to the bleaching effect in these regions, which ranges from 0.37 days (southeast Asia) to 2.09 days (Arctic)."

**5.** *In summary, maybe the authors could list what are the major uncertainties in their analysis of BrC radiative impacts and what is needed to address them.*
Reply:
We address the main uncertainties in the model parameterizations, the model analysis, and the uncertainties in the vertical distribution of aerosols (addressed in comment 2).

We made a change on **page 6, line 1** - "The GFED 3.1 emissions were used in this study to allow for direct comparison between this study and Jiang et al. (2016). The more recent GFED 4 emission dataset shows an 11% global increase in fire emissions from

GFED 3.1 (Werf et al., 2017), which may result in a slightly stronger climate impact from biomass burning aerosols than that shown in this study."

We made a change on **page 6, line 25** - "Uncertainty in $k_{OA}$ from this parameterization is associated with the lab measurements of the particle mass, the range in assumed complex refractive index for BC, the mixing state of BC and OA, the measured real part of the OA refractive index, and the measured absorption coefficients used in optical closure calculations (Saleh et al., 2014)."

We made a change on **page 7, line 12** - "A few assumptions in this model simulation introduce uncertainty in the representation of BrC in CESM. One of those assumptions is neglecting absorption by BB SOA (Lin et al., 2014; Saleh et al., 2015) or absorbing aromatic SOA (Wang X. et al., 2014; Jo et al., 2016; Wang X. et al., 2018), which is neglected due to the lack of SOA speciation in the model. This assumption, in conjunction with the use of GFED 3.1 instead of GFED 4, may act to underestimate the climate effect due to BrC. Another assumption is the model use of a volume mixing assumption, which may act to overestimate aerosol light absorption (Jacobson, 2000; Adachi et al., 2011). We also assume that the BC-to-OA ratio in transported smoke is similar to BC-to-OA from the source region, allowing for the use of a BC-to-OA ratio at each gridcell at every time step to calculate $k_{OA}$ in each gridcell. The uncertainty in $k_{OA}$ associated with this assumption is small (<10% for BB emissions assuming transport from the Equator to the Arctic (not shown)) and is assumed to be negligible.

We made a change on **page 8, line 11** - "While the parameterization depends on OH concentration in the atmosphere, by matching the BrC lifetime to observations the parameterization also includes photochemical oxidation and other bleaching effects that may have been active in the observed smoke plumes. This is true of the regions in which the observations were taken, but may not hold true for global sites or seasons with lower insolation. Uncertainty in this parameterization is associated with the low availability of observational data, and could be improved with more field measurements of BB smoke aging at different latitudes."

*Minor comments.*

**6.** *P2, L12: Feng et al did not consider BrC bleaching, so this is likely a large over estimation, which should be noted.*
Reply:
We make it clear later in the paper that the BrC estimations by earlier modeling studies are overestimated. This particular line is referring to the burden of OA compared to BC, and so we left it as is.

**7.** *Don't really understand the layout of the first 3 equations. Eq 1 should be something like RI = ...*

Reply:
Equation (1) has been modified to

$$RI = 1.7(\pm0.2) + k_{OA}i = 1.7(\pm0.2) + k_{OA,550}\left(\frac{550}{\lambda}\right)^w i$$

**8.** *P9 L18, typo BRC_CL ??*
Reply:
Fixed.

**9.** *Fig 6 and associated discussion and in the sections that follow; be specific about the brown carbon included in the model, ie, was it BRC, CRC_CNST or BRC_BL?*
Reply:
We have paid better attention to how figures and discussion are worded so that the reader can better understand which model simulation is being used.

**10.** *The model considers BrC bleaching just due to OH. Is this the only route for bleaching? Please justify. What are the limitations with this assumption?*
Reply:
Due to the fact that the BrC bleaching parameterization was designed to match observed BrC lifetimes, inherent in its timing is the inclusion of photochemical oxidation and other effects contributing to BrC aging. However, this is regionally specific to the location of the observations and we discuss some ways the parameterization could be improved (see comment 5).

Reply:
We include a table of comparison between three different AAE surface measurements: Ascension Island observations from the LASIC (Layered Atlantic Smoke Interactions With Clouds (LASIC) campaign, the Las Conchas fire in NM, and biomass burning and prescribed burns from SEAC$^4$RS (Studies of Emissions and Atmospheric Composition, Clouds and Climate Coupling by Regional Surveys). We also mention biomass burning AAE from three 2018 fires in NM.

We added **Table 2.**

[revised manuscript text omitted]